# The SpinBus architecture for scaling spin qubits with electron shuttling

Matthias Künne [1,3], Alexander Willmes [1,3], Max Oberländer[1], Christian Gorjaew[1], Julian D. Teske [1], Harsh Bhardwaj [1], Max Beer [1], Eugen Kammerloher[1], René Otten [1,2], Inga Seidler [1], Ran Xue [1], Lars R. Schreiber[1,2] ✉ & Hendrik Bluhm [1,2] ✉

Quantum processor architectures must enable scaling to large qubit numbers while providing two-dimensional qubit connectivity and exquisite operation fidelities. For microwave-controlled semiconductor spin qubits, dense arrays have made considerable progress, but are still limited in size by wiring fan-out and exhibit significant crosstalk between qubits. To overcome these limitations, we introduce the SpinBus architecture, which uses electron shuttling to connect qubits and features low operating frequencies and enhanced qubit coherence. Device simulations for all relevant operations in the Si/SiGe platform validate the feasibility with established semiconductor patterning technology and operation fidelities exceeding 99.9%. Control using room temperature instruments can plausibly support at least 144 qubits, but much larger numbers are conceivable with cryogenic control circuits. Building on the theoretical feasibility of high-fidelity spin-coherent electron shuttling as key enabling factor, the SpinBus architecture may be the basis for a spin-based quantum processor that meets the scalability requirements for practical quantum computing.

The prospect of noisy intermediate-scale quantum (NISQ) computing raises high expectations. However, it is likely that a significant part of the foreseen applications will only be accessible via quantum error correction to mitigate errors caused by noise, spurious coupling, and crosstalk[1]. The resulting overhead leads to a need for millions of physical qubits, which requires highly nontrivial advances compared to today's devices. Electron-spin qubits in semiconductor quantum dots have the unique feature of being directly compatible with industrial CMOS processing[2]. At the level of few-qubit devices, all-electrical operation of single- and two-qubit gates above the error correction threshold have been demonstrated[3–11]. Furthermore, the operation of multi-qubit devices has been shown in several material systems[12–15]. Building on these promising results, the immediate next challenge for semiconductor qubits is scaling-up in two dimensions while simultaneously maintaining high operation fidelities to realize qubit numbers

needed for NISQ computing, i.e., on the order of 100 qubits. To fully deliver on the promises associated with CMOS compatibility, a way to scale up to millions of qubits must be found. A key challenge at the quantum layer is the short range (≈100 nm) of the exchange interaction typically used for high-fidelity two-qubit gate operations. Architectures based on direct coupling thus lead to crowding of gate electrodes and their wiring[2,16], referred to as the wiring fan-out problem, as well as significant inter-qubit crosstalk[17].

To address these challenges, dense qubit arrays using crossbar network addressing schemes with reduced wiring density, as well as sparse arrays of qubits with integrated classical electronics at cryogenic temperatures, have been proposed[18]. Dense architectures based on crossbar addressing schemes typically apply the same control pulse to many qubits and thus require a challenging level of qubit homogeneity[19]. Tuning the qubit properties with local

[1]JARA-FIT Institute for Quantum Information, Forschungszentrum Jülich GmbH and RWTH Aachen University, 52074 Aachen, Germany. [2]ARQUE Systems GmbH, 52074 Aachen, Germany. [3]These authors contributed equally: Matthias Künne, Alexander Willmes. ✉e-mail: lars.schreiber@physik.rwth-aachen.de; bluhm@physik.rwth-aachen.de

transistor-based circuits can somewhat ameliorate this issue, but imposes demands on transistor and capacitor size[20] that are well beyond current capabilities. As an alternative path to avoiding these difficulties, we propose a concrete realization of the quantum layer that is based on readily available technology. A key element is the use of electron shuttling to form a sparse qubit array with sufficient space for wiring in near-term implementations and local control electronics with a footprint commensurate with the qubit density for very large qubit numbers in the longer term. This detailing of the quantum-level architecture complements the proposal for scaling such a shuttling-based sparse array using cryoelectronic control circuits[16]. The use of electron shuttling, i.e., moving electrons between sites where qubits are manipulated, enables local exchange-based two-qubit gates without requiring a dense qubit array. Gate-based electron shuttling has been realized in both GaAs/(Al,Ga)As and Si/SiGe. By implementing Landau-Zener transitions between adjacent quantum dots in the so-called bucket brigade mode, the transport of single electrons and coherent transfer of electron spins has already been demonstrated[21–26]. Recently, single-electron transport by so-called conveyor-mode shuttling was shown[27], where a quantum dot used to trap the qubit is continuously translated to distant qubit sites, requiring a length-independent number of wires and also less tuning. In a 10-µm-long prototype device, charge shuttling in one direction and back across a distance of 19 µm with a fidelity of 99.7% has been achieved[28].

The concept and feasibility of coherent conveyor-mode electron shuttling was analyzed in detail by ref. 29. The confinement potential is chosen much stronger than the background disorder potential, targeting an adiabatic motion that leaves the electron in the orbital ground state. With a shuttling velocity of a few m/s, electrons can be transferred fast enough to limit spin dephasing due to $T_2^*$-effects such as charge and hyperfine noise. However, nonadiabatic transitions between different valley and potentially orbital states set an upper bound on the velocity. For a minimal valley splitting of 20 µeV, a coherent transfer with an error rate below $10^{-3}$ over a distance of 10 µm is predicted for a shuttling velocity of $v = 8$ m/s, which we assume throughout this paper. A subsequent study[30], as well as the first experiments[31], show that occasional lower values of the valley splitting can be avoided by laterally shifting the trajectory of the shuttled electron. In a 1 µm Si/SiGe prototype device with a natural abundance of Si isotopes (similar to ref. 27), spin-coherent shuttling with a maximum velocity of 2.8 m/s across an accumulated distance of at least 2.4 µm has been demonstrated. The spin dephasing time of the shuttled electron spin is enhanced by motional narrowing, which contributes even in the absence of $^{29}$Si isotopes due to remaining $^{73}$Ge isotopes, and leads to a fidelity of ~99% for the transfer of a spin quantum state over a nominal shuttling distance of 560 nm[32]. In addition, motional narrowing is also expected for charge noise, albeit with a longer correlation length set roughly by the distance of the noise source from the channel[29].

## Results

### The SpinBus architecture and its elements

In this manuscript, we present the SpinBus architecture, which leverages the conveyor-mode shuttling device named Quantum Bus (QuBus) as used in demonstration experiments[27,28,32] to connect qubits (Fig. 1a). Like established semiconductor qubit devices, the QuBus device employs a stack of electrostatic gates on top of a Si/SiGe heterostructure that confines electrons in the z-direction (Fig. 1b). Lateral screening gates define a one-dimensional channel in the xy-plane, while clavier gates placed above are used to generate moving quantum dots. Every fourth clavier gate is electrically connected, thus eliminating the need for fanning out each individual gate. Four phase-shifted sinusoidal signals $V_i$, $i = 1 \dots 4$, applied to the resulting four sets of clavier gates enable a continuous translation of the quantum dots.

The signals $V_i$ have the form[27]

$$V_i = A_S \cos(\varphi(t) - \Delta\varphi_i). \qquad (1)$$

Here, $A_S$ ~100 mV is the signal amplitude and $\varphi(t) = 2\pi f \cdot t$ with frequency $f$ is the phase with phase offset $\Delta\varphi_i = \pi/2(i-1)$. Hence, the number of required signals is independent of the distance between qubit sites. A DC bias relative to the $V_i$ can be applied to Ohmic contacts to adjust the chemical potential. Lateral shifts of the shuttling path to avoid critical regions (see Supplementary Note 1) can be achieved by antisymmetric changes of the voltages on the screening gates synchronized with the electron motion[30,31].

Based on the QuBus component as a coherent link, we propose a layout of tileable unit cells as building blocks for the quantum layer of the SpinBus architecture (Fig. 1c). The unit cell (Fig. 1d) provides the means for initializing, reading-out and performing gate operations in two specialized zones, i.e., the initialization and readout (IR) and the manipulation zone. Shuttling lanes connect both the operational zones and adjacent unit cells. We anticipate that the length of the shuttling lanes in the order of 10 µm will reflect a reasonable trade-off between shuttling-induced errors and time versus space for wiring and local electronics[33]. The spatial separation between different manipulation zones and qubits avoids unwanted inter-qubit coupling and helps to address qubits individually, thus avoiding control crosstalk errors. This comes at the cost of shuttling errors, which add to the errors of locally executed gates.

The QuBus geometry is based on the recent demonstration experiments of conveyor-mode shuttling, where a separation of the screening gates by 200 nm, a gate width of 62 nm, and a gate pitch of 70 nm have been used[27]. For the validation of the gate layouts with electrostatic finite-element-method (FEM) models (see Methods section "Electrostatic simulations and orbital splitting"), we chose a slightly larger gate pitch of 100 nm, including a global top gate that can be biased with a separate voltage $V_{tg}$ ~100 mV. For the operation of some elements, micromagnets are placed in suitable locations approximately 150 nm above the quantum well. For magnetostatic modeling (see Methods section "Micromagnet design"), we assumed an external in-plane magnetic field $B_{ext} = 20–50$ mT in the y-direction.

Two-dimensional connectivity is implemented by a three-way T-junction connecting two perpendicular shuttling lanes (Fig. 2a) without requiring any additional gates. Compared to a four-way junction[16], gate crowding is reduced and potential shaping simplified. The two supported operations are qubit motion in a straight line (straight shuttling) and around the corner (corner shuttling). Straight shuttling is implemented analogously to normal conveyor-mode operation, with the voltages on the perpendicular branch being constant. Due to the rapid decay of electric fields, crosstalk is avoided by storing qubits in the perpendicular branch at least 100 nm away from the junction when operating the straight branch. For corner shuttling, a quantum dot initially moving along the straight branch is stopped at the intersection and then transferred into the perpendicular branch. Figure 2b shows the corresponding potentials for different points in time during the adiabatic transfer using appropriately adjusted voltage pulses. Selected line cuts of the potential and the time evolution of the shuttling phases are presented in Supplementary Figs. 1, 2, respectively. For both operations, the transport direction can be inverted by reversing the shuttling pulses. For coherent shuttling, the electron motion should reflect a smooth translation of the potential, rather than tunneling between disorder-induced stationary quantum dots. A useful metric for this requirement is the orbital splitting for the moving quantum dot containing the qubit. Langrock, Krzywda et al. determined that the required confinement strength to safely prevent the splitting of the shuttled potential minimum into a double quantum dot configuration in the presence of ensembles of randomly distributed charged defects at the Si/SiO$_2$ interface corresponds to an orbital

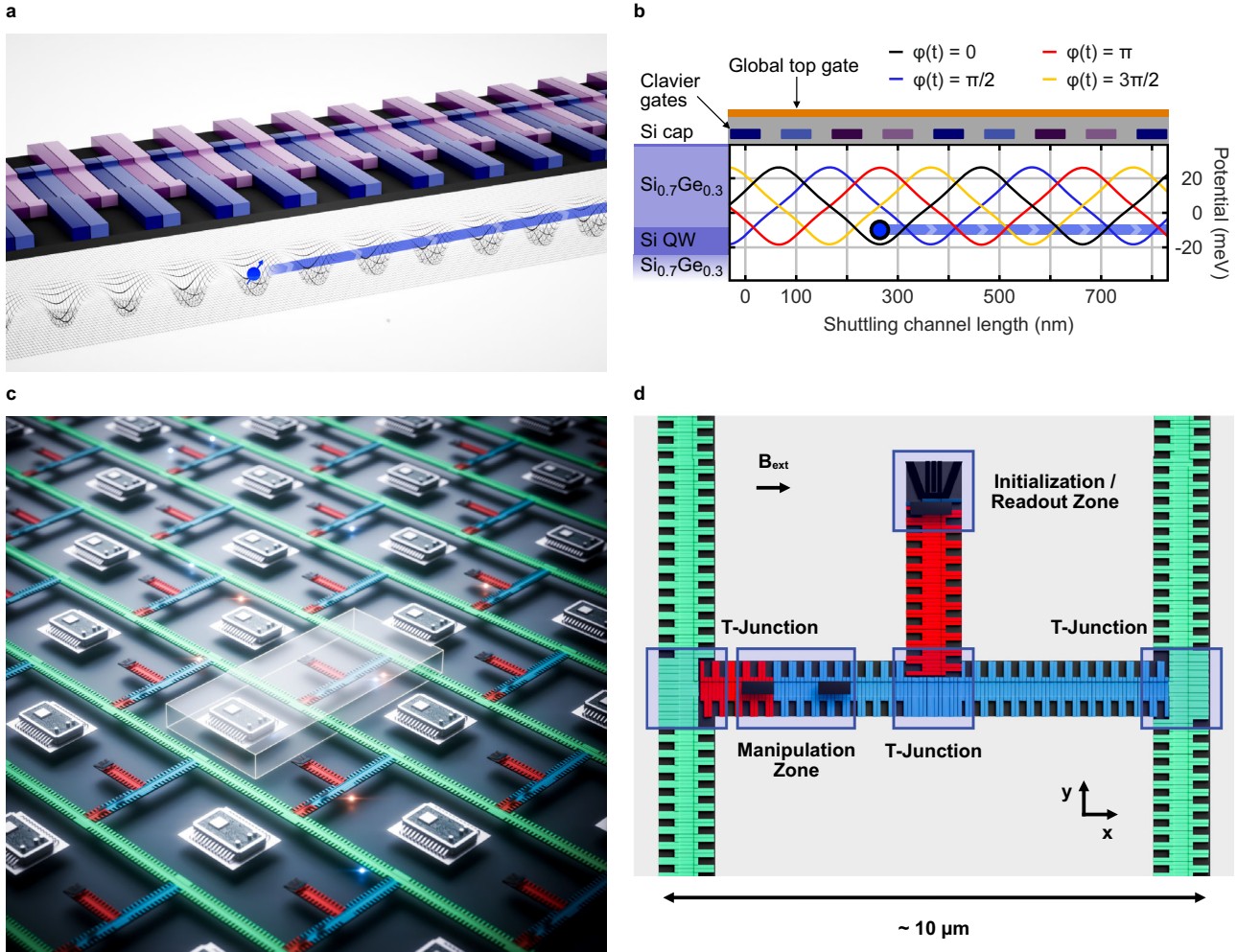

**Fig. 1 | Layout and operation of the QuBus device as a building block for the SpinBus architecture. a** 3D visualization of the QuBus device consisting of two lateral screening gates defining a 1D electron channel and periodically connected clavier gates. The four gate sets connected to different control signals $V_i$ are color-coded. **b** Schematic of the Si/SiGe heterostructure providing the quantum well (QW). Line cuts of the traveling potential generated by the gate stack are depicted for four different phases $\varphi(t)$. The occupied potential minimum is indicated by a blue circle. The gate stack is depicted above the potential line cuts. **c** The quantum processor chip consists of unit cells tiled like a brick wall. One unit cell is highlighted, the heterostructure is visualized transparently in most areas, and local electronic components are shown symbolically. Unit cells are connected via the green-colored shuttling lanes controlled by a signal set shared across unit cells. Red and blue-colored shuttling lanes are controlled individually in each unit cell. **d** A unit cell consists of three T-junctions for 2D connectivity, an initialization and readout zone, and a manipulation zone. We expect a spatial extent of unit cells in the order of 10 μm.

splitting of at least 1 meV. This criterion is in agreement with experimentally obtained values typically found in static quantum dots[34]. During straight shuttling, the orbital splitting equals or exceeds the threshold at all times (Fig. 2c). A drop in the orbital splitting during corner shuttling caused by the asymmetry of the gate layout at the junction which reduces confinement can safely be prevented (Fig. 2d) by dynamically pulsing the outer screening gate of the straight branch during transfer (Supplementary Fig. 2). The pulse pushes the electron towards the branching channel, reduces the effect of the asymmetry and thus increases the confinement. To avoid any influence on other qubits stored in the straight branch, a segmentation of the outer screening gate at the junction can allow a local pulsing.

The initialization and readout (IR) zone consists of a single-electron transistor (SET) tunnel-coupled to a shuttling lane, thus enabling loading and detecting charges (Fig. 3a). Ohmic contacts on both sides of the SET provide source and drain reservoirs, and electrons are injected into the shuttling lane via the SET. Besides one plunger and two barrier gates for the SET, we propose two additional individually contacted gates at the beginning of the shuttling lane (Fig. 3b). The first controls the tunnel barrier to the SET, and the

second the potential of the first quantum dot in the QuBus channel. A second moving quantum dot can be controlled independently by the four sets of clavier gates. For qubit initialization and readout, Pauli spin blockade (PSB) in the resulting double quantum dot is utilized to enable simpler and faster readout discrimination than, e.g., spin-selective tunneling[35,36]. The required parallel magnetic field gradient $\partial B_\parallel$ is generated by a micromagnet placed directly above the shuttling lane adjacent to the SET. The initialization sequence follows standard procedures and is presented in Fig. 3c. It starts with loading two electrons into a first quantum dot (step I), forming a tunnel-coupled double quantum dot configuration (step II) while the second quantum dot is kept at a sufficiently higher potential during the adjustment of the inter-dot tunnel barrier (step III) to remain in a S(2,0) state. Sweeping the detuning $\epsilon$ transfers the S(2,0) state to a (1,1) configuration, where the gradient magnetic field splits the $T_0$ and S(1,1) into $|\uparrow\downarrow\rangle$ and $|\downarrow\uparrow\rangle$ (step IV). Thus, the $|\downarrow\uparrow\rangle$ will be occupied if the detuning is pulsed adiabatically with respect to orbital, spin, and valley excitations, but including a short diabatic sweep over the ST_-crossing. Lastly, the spin-up state is shuttled away to be used as a qubit. The spin-down electron can be kept in the first quantum dot as a reference spin

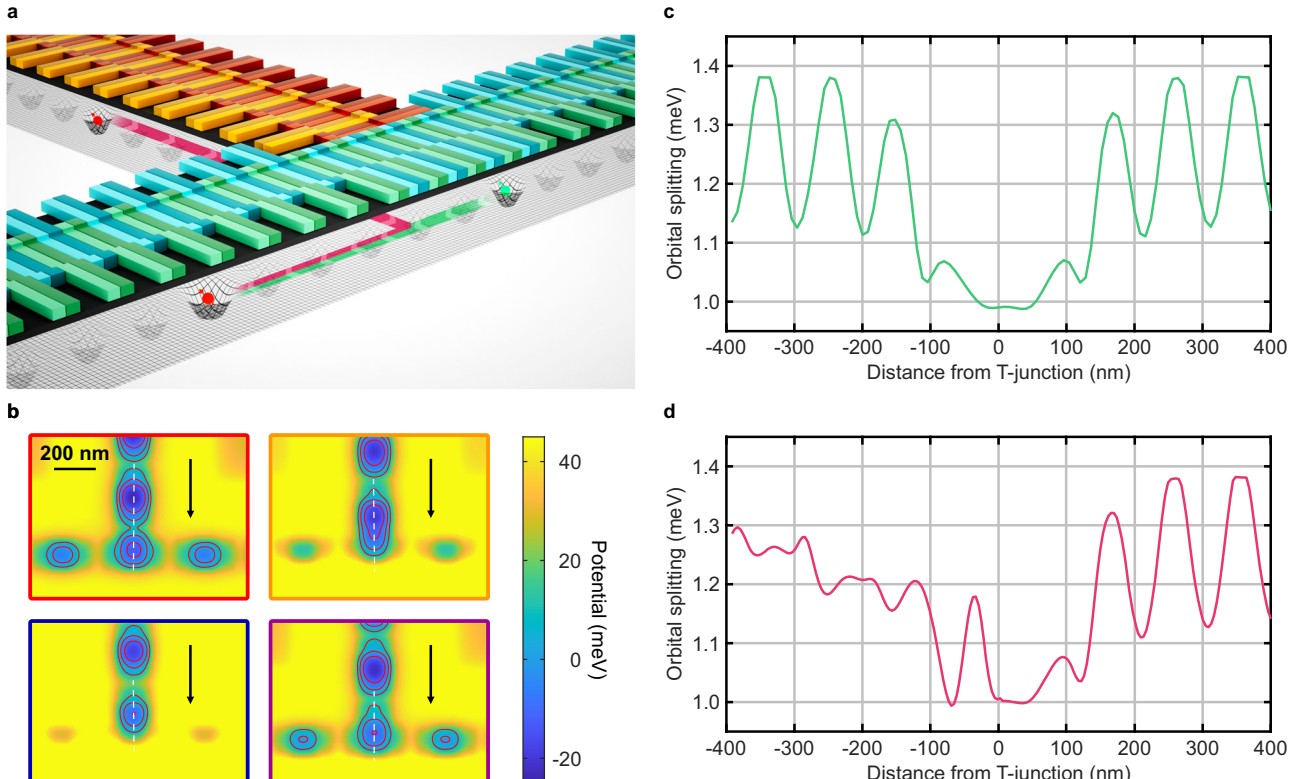

**Fig. 2 | Layout and operation of the T-junction. a** 3D visualization of the T-junction consisting of two perpendicularly joined QuBus elements. Straight shuttling (red path) and corner shuttling (green path) are shown. **b** 2D potential at the T-junction for different points in time during corner shuttling. Arrows indicate the shuttling direction, and white dashed lines indicate the positions and lengths of the line cuts in Supplementary Fig. 1. **c** The orbital splitting during straight shuttling is always sufficiently large. **d** During corner shuttling, dynamically adjusting the screening gate voltage ensures an orbital splitting within the target range.

state for later readout. The corresponding time traces for the shuttling phase $\varphi(t)$ and detuning are shown in Supplementary Fig. 3. To implement readout, the initialization sequence is reversed, and PSB is employed to determine the qubit's state. Any established method for SET readout can be used, though we speculate that baseband readout with cryogenic transistors[37–39] will yield the best performance-complexity trade-off.

Single- and two-qubit gate operations are performed in the manipulation zone, which is formed by joining two shuttling lanes (Fig. 4a). Two independent QuBus elements enable sufficient control over both detuning and tunnel coupling of a double quantum dot potential formed at the junction, thus eliminating the need for additional separately contacted gates. Two micromagnets provide the necessary magnetic field gradients (Fig. 4b). For single- and two-qubit gates, a micromagnet is placed off-center from the junction above one QuBus element. On the other side of the junction, an additional micromagnet for single-qubit gates is located above the other QuBus element at a sufficient distance to avoid compromising the longitudinal field gradient at the junction. Thus, the manipulation zone allows performing single-qubit gates on two qubits independently.

Single-qubit gates are implemented by electric-dipole spin resonance (EDSR), in which an effective oscillatory transverse magnetic field for driving Rabi oscillations is generated by displacing the electron in a perpendicular magnetic field gradient. Unlike conventional EDSR manipulation, where the electron position oscillates typically up to one nanometer[3], we propose a shuttling-mode EDSR building on the capability of moving the electron over arbitrary distances. For high fidelities, we estimate an oscillation amplitude in the order of 10 nm to be a good choice. The larger amplitude allows the use of significantly weaker magnetic field gradients, which reduces the sensitivity to charge noise. While the influence of spin-orbit interaction (SOI) during

shuttling has been found to be minor[29], it can safely be neglected for manipulation as the synthetic SOI used for EDSR is normally dominant. Regarding crosstalk, driving a single-qubit gate on a qubit right of the junction in Fig. 4b causes a relative electrostatic shift corresponding to 0.5% of the driving amplitude for the other qubit located left of the junction (orange circle), 250 nm from the driven shuttling element. Conservatively assuming the same resonance frequency, this translates to an infidelity of approximately $6 \times 10^{-5}$ for a $\pi$-gate. For the more distant qubit in the right single-qubit manipulation region, crosstalk is even weaker. In addition, the remaining crosstalk can be reduced further by specifically tailored pulses accounting for the respective opposite single-qubit operation.

For electron-spin qubit platforms utilizing micromagnets, the natural choice for the implementation of CNOT-like two-qubit gates (see Methods section "CNOT gate synthesis") is the controlled-phase (CPHASE) gate based on the exchange interaction $J(t)$ between two tunnel-coupled quantum dots[40–42], which is switched adiabatically with respect to a Zeeman energy difference $\Delta E_Z$ between the two quantum dots. This configuration is achieved by shuttling both electrons to the junction at the center of the manipulation zone (Fig. 4c) with pulses as shown in Supplementary Fig. 4 while maintaining zero detuning. The control of the exchange coupling via the inter-qubit distance while maintaining zero detuning essentially amounts to barrier control, which features a lower charge noise sensitivity compared to controlling the exchange interaction via the detuning[5,43]. Figure 4c shows the simulated potentials during the formation of a double quantum dot. The separation and barrier height during the two qubit gates are similar as in conventional quantum dot structures, thus validating the robustness of the procedure with respect to disorder. The absence of tunnel coupling to other sites further increases this robustness in comparison to arrays with multiple quantum dots.

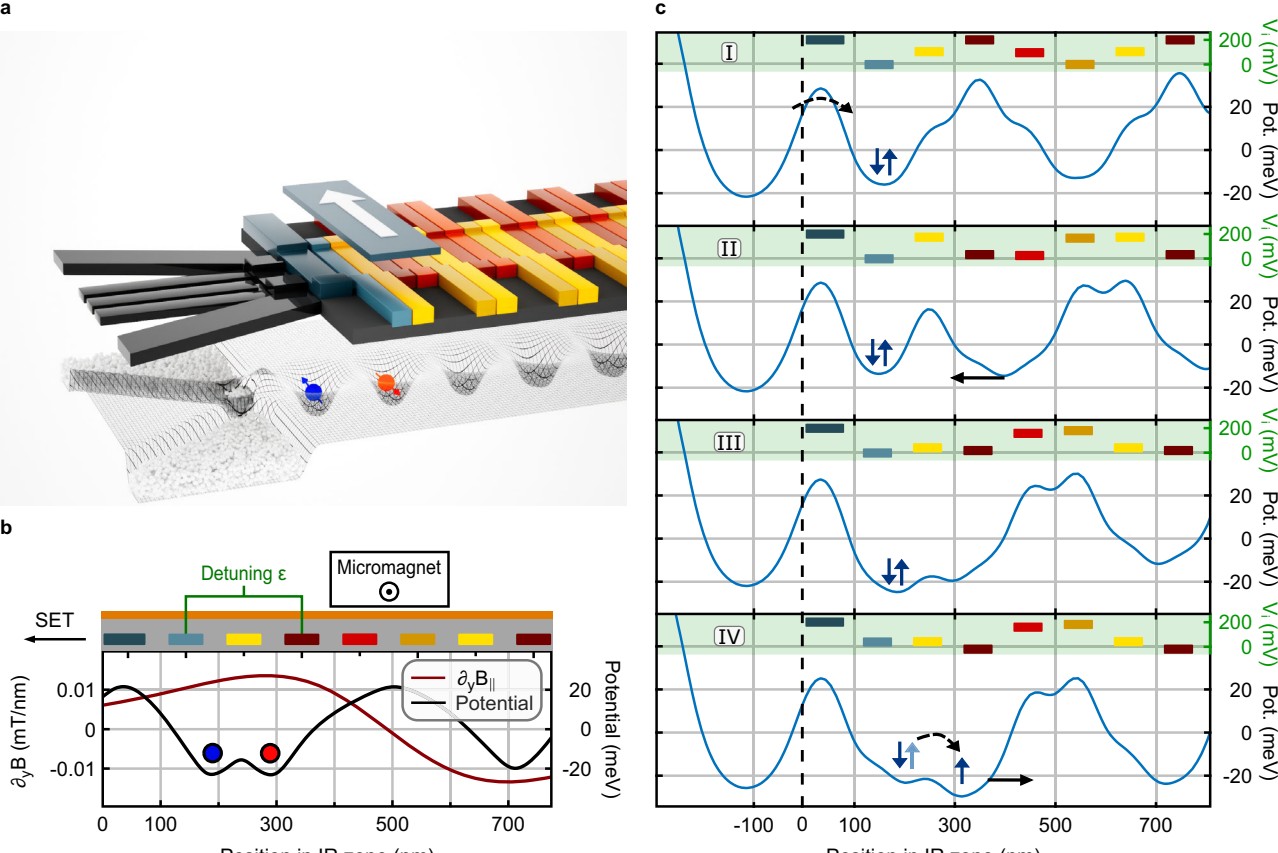

**Fig. 3 | Layout and operation of the initialization and readout (IR) zone. a** 3D visualization of the IR zone consisting of a QuBus element adjacent to an SET, and a micromagnet. Two gates next to four sets of clavier gates are individually controlled. **b** Cross-section including the gate layout showing the schematic double quantum dot potential and simulated magnetic field gradient $\partial_y B_\parallel$ along the shuttling channel. Red and blue circles represent the positions of two electrons in a double quantum dot configuration. **c** Potential line cuts while initializing a qubit using PSB (blue arrows represent the electrons' spin states). The color-coded bars correspond to the gates from panels **a** and **b**, and their vertical positions indicate the applied voltage $V_i$. Tunneling is indicated by dashed black arrows and solid black arrows mark the translation of quantum dots. Step I: loading of a S(2,0) state from the SET into the first quantum dot. Step II: moving a second quantum dot close to the first quantum dot. Step III: detuned double quantum dot. Step IV: applying a detuning sweep to transfer S(2,0) to $|\uparrow\downarrow\rangle$ followed by a shuttling pulse to inject the qubit into the shuttling channel.

## Fidelity of quantum operations

To estimate the achievable performance, we simulated the dynamics of each quantum operation using the simulation package qopt[44], including optimization of the control pulses (see Methods section "Operation fidelities"). The fidelities were computed based on a noise model including quasistatic nuclear spin noise affecting the Zeeman splitting as well as quasistatic and white charge noise with amplitudes extracted from past experiments[3,45,46]. With appropriate calibration, the combination of quasistatic and white charge noise can serve as a conservative proxy for 1/f-noise typically found in real devices. We included coupling of the charge noise to the qubit via the detuning affecting the exchange coupling as well as via position fluctuations. The latter affects the single spin dynamics due to the magnetic field gradient as well as the exchange coupling at zero detuning. This noise model covers the effects we consider as experimentally most relevant and was shown to be in good agreement with experimental results[47]. For the initialization and readout procedure, we identified fast charge noise as the main limiting factor and obtained fidelities above 99.9% if parasitic inter-dot orthogonal magnetic field gradients remain sufficiently small (see Methods section "Micromagnet design"). To evaluate single-qubit operations, we applied a sinusoidal shuttling EDSR-pulse in resonance with the Zeeman splitting to a qubit model with spin and valley degree of freedom. We identified fast charge noise causing position fluctuations as the dominating noise contribution and find fidelities exceeding 99.9% as long as the valley splitting is greater than 30 μeV and exhibits integrated variations of less than 100 μeV along the path. For two-qubit gates, the relevant infidelity contribution arises from quasistatic position noise affecting the exchange interaction, and we obtain a fidelity of 99.9%.

## Operating concept and system complexity

The two-dimensional array of the architecture is well suited for the implementation of surface codes, which can be considered the mainstream concept for quantum error correction[1], as well as NISQ algorithms. As an exemplary operation, we show the elementary surface code gate sequence in Fig. 5, requiring a square array of qubits with nearest-neighbor coupling. Every second qubit serves as a data qubit storing quantum information, and every other one as an ancilla qubit, each detecting one of two possible types of errors called $\hat{X}$ and $\hat{Z}$ stabilizers[48]. As each manipulation zone can simultaneously operate two qubits, each unit cell is identified with one data qubit highlighted in blue and one adjacent ancilla qubit highlighted in green and yellow, respectively (Fig. 5a). An error detection cycle consists of initializing the ancilla qubits, CNOT gates with the four adjacent data qubits, which we choose as stationary, and subsequent readout of the ancilla qubits. Realizing such a cycle in the SpinBus architecture requires the shuttling of ancilla qubits to and from different manipulation zones between local gate operations (Fig. 5b).

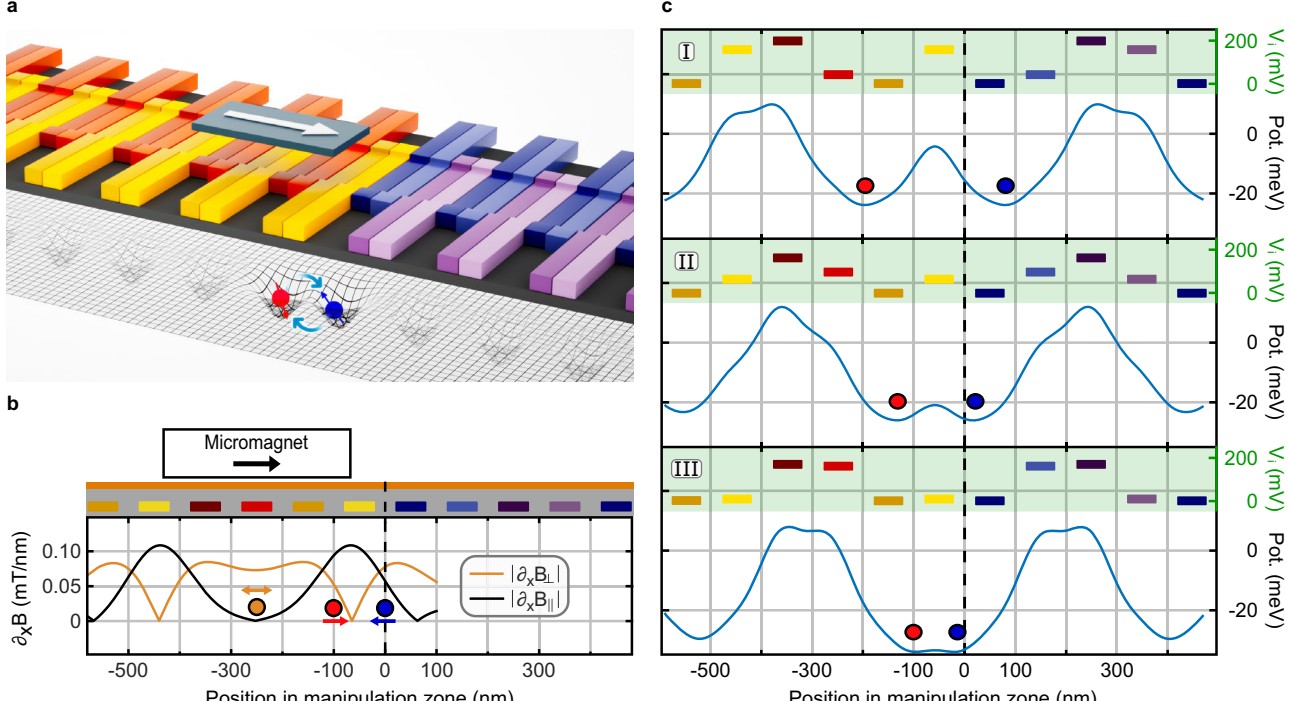

**Fig. 4 | Layout and operation of the manipulation zone. a** 3D visualization of the manipulation zone consisting of two joined QuBus elements and two micromagnets. Shown is an exemplary two-qubit operation. **b** Cross-section including the gate layout showing the required magnetic field gradients for single- and two-qubit gates along the manipulation zone. The orange circle shows the position of the qubit during the single-qubit gate operation and is driven periodically in the region of a large perpendicular magnetic field gradient $|\partial_x B_\perp|$. The red and blue circles indicate the positions of the qubits during the two-qubit gate operations, respectively. Both are pushed together at the location of a large parallel magnetic field gradient $|\partial_x B_\parallel|$. **c** Potential line cuts showing the smooth formation of a tunnel-coupled double quantum dot potential appropriate for two-qubit operations as the two translated quantum dots approach the center of the manipulation zone. The color-coded bars correspond to the gates from panels **a** and **b**, and their vertical positions indicate the applied voltage $V_i$.

The predicted fidelities are in the range of what is needed to achieve a reasonable, logical qubit performance and overhead. Thus, individual logical qubits are within reach for qubit numbers that can be realized with conventional control and packaging approaches. Since integrating on the order of 100 qubits approaches the limits of connecting room temperature control, we first present a near-term implementation with 144 qubits based on realistic assumptions regarding qubit homogeneity, control electronics, and cooling hardware. The estimate of the number of required signals is based on an economical operating strategy detailed in Supplementary Note 2. We then discuss a concrete scaling perspective to much larger qubit numbers by using cryoelectronic control circuits, which significantly reduce the number of required external control lines.

Considering shuttling signals (also used for qubit control) and additional local AC and DC signals of the IR zone and the screening gates, a quantum processor chip with $N$ unit cells requires $15N + 4$ AC and $3N + 4$ DC signals. While there are no inherent scaling limitations to our architecture at the quantum layer, the wiring requirements have to be compatible with cryostat wiring, packaging, and back-end-of-line (BEOL) technology. We estimate that currently available wiring solutions in cryostats of about 1000 coaxial cables[49] are the most limiting factor and can accommodate a quantum processor chip with $9 \times 8 = 72$ unit cells. This corresponds to 144 simultaneously operable qubits if two qubits per unit cell are loaded. Storing additional qubits away from manipulation zones can further increase the qubit number.

## Scaling perspective using cryoelectronic control circuits

For large-scale quantum computing with many error-corrected qubits, however, conventional control and packaging approaches are less appealing. Here, integrated control solutions offer an attractive pathway. The SpinBus architecture features good prospects in this respect because the purely capacitive impedance of control electrodes, low operating frequency, and robust coherence of spin qubits facilitate the use of cryogenic CMOS control circuits. The variable unit cell size can be adjusted to the required size of dedicated control circuits for each unit cell, so that direct wiring, e.g., via flip-chip bonding, can eliminate the wiring fan-out problem. First, estimates of the size of control circuits for spin qubits lead to values in a compatible range[16,33]. Next to the size of control circuits, their power dissipation will be a concern in the light of limited cooling power at low temperature. For DC bias, a consumption at a level of a few nW per channel has already been shown[50]. While we favor a qubit temperature on the order of 100 mK to ensure a minimal loss of gate fidelity, thermally isolating flip-chip solutions may allow the operation of electronics at a higher temperature than the qubits. Working around 2 K would potentially make cooling powers at the level of Watts accessible. For this purpose, we propose the implementation of thermal insulation of the quantum layer from the electronics by a broadband phononic Bragg reflector[51] to sustain a temperature gradient over a high-density interconnect solution. Simulations[52] indicate that a heat load below 1 mW/cm² can be achieved with a thickness compatible with high-density vias with a micron-scale pitch. Using superconductors such as NbN or NbTiN with a critical temperature of a multiple of the operating temperature can lead to a very small heat transfer through the vias. While it remains to be seen if the dynamic qubit and shuttling control signals can be generated within the resulting power budget, qubit control can also be implemented by multiplexing of externally generated pulses[53], for example, using simple cryo-CMOS switches[54,55]. As a reverse approach to adapting pulses to individual qubits, the qubit response could be tuned to these fixed pulses using DC gate voltages. While more demanding and arguably less elegant than global crossbar addressing, this approach has the advantage of not making

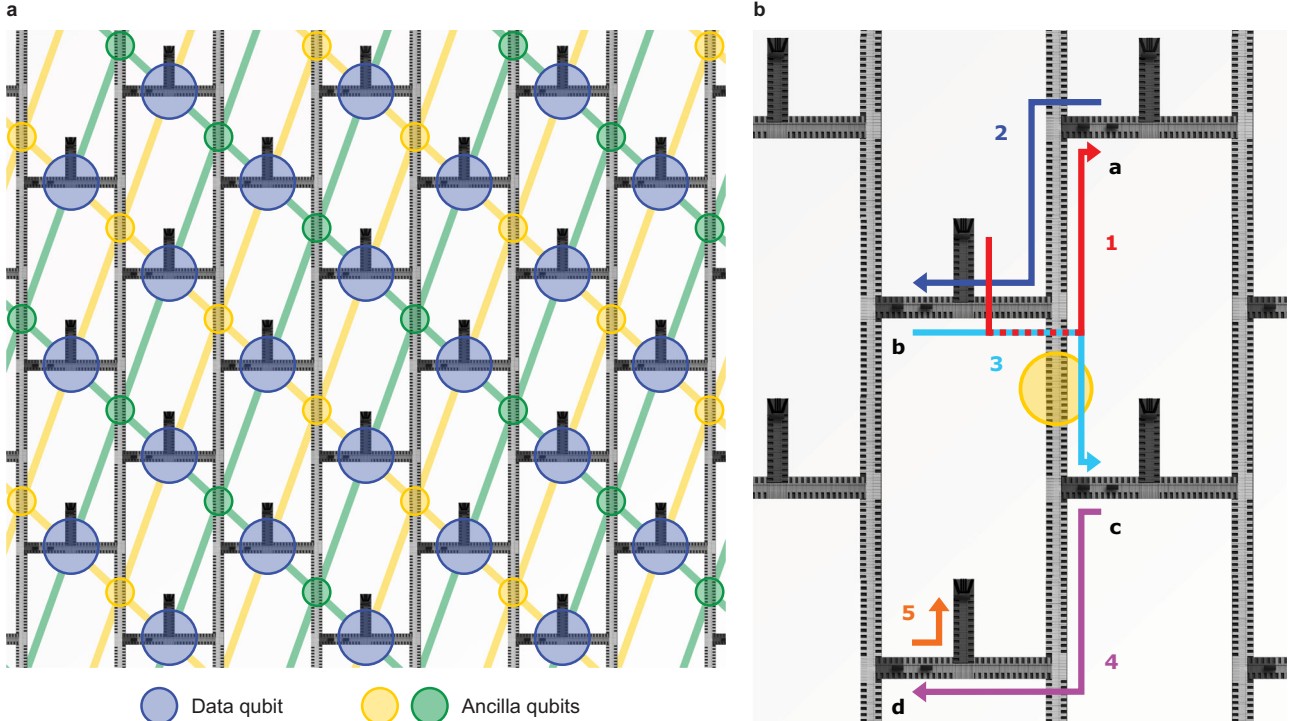

**Fig. 5 | Exemplary surface code implementation in the SpinBus architecture. a** Mapping of data and ancilla qubits to unit cells. Each diagonal line segment represents a required qubit interaction. **b** Shuttling paths for an ancilla qubit in order to implement an $\hat{X}$ stabilizer. Associated data qubits are designated with **a**–**d**. Arrows mark the shuttling paths to all involved manipulation zones, whereas the numbers indicate the order of operations. Steps 1 and 5 include the initialization and readout of the ancilla qubit, respectively.

assumptions regarding the homogeneity of the qubit parameters. For readout, heterojunction-bipolar-transistors (HBTs) allow single-shot readout in less than 10 μs at powers below 800 nW[39], which likely can be reduced with optimized sensor designs[56,57].

## Discussion

In summary, we have detailed a concept to leverage electron shuttling for the realization of a semiconductor-based quantum processor with 2D coupling, as required for quantum error correction based on the surface code. The proposed layout can be optimized for other use cases or according to a trade-off between shuttling and gate errors. For example, it was found that for parity encoding of quantum approximate optimization algorithms, four rather than two single-qubit manipulation zones promise a very good performance[58]. To validate the feasibility, we performed electrostatic simulations for all device layouts and modes of operation. For the estimation of operation fidelities, we used realistic noise models and obtained fidelities for single- and two-qubit gates exceeding 99.9%. The fabrication is possible with present-day industrial semiconductor processing. Furthermore, the architecture is compatible with established packaging and wiring techniques such as BEOL via fabrication and flip-chip bonding. While we considered an implementation in Si/SiGe, the SpinBus architecture can potentially be transferred to other types of gate-defined semiconductor qubits.

Our architecture proposal features a number of strengths, but it clearly hinges on the theoretically predicted feasibility of spin-coherent electron shuttling. While the first experiments on spin-coherent transport are promising, an implementation with high fidelity and mitigating low values of the valley splitting in Si/SiGe (see Supplementary Note 1 for details) will be an essential next step. Reaching the projected fidelities and required yield could quickly put semiconductor qubits on the map for NISQ-type quantum computing. The combination with cryoelectronic control systems, which is facilitated by the variable qubit spacing, robust coherence of semiconductor qubits, the purely capacitive load of gate electrodes and the relatively low operating frequency could carry to much larger systems, eventually enabling error-corrected quantum computing. Recent advances in cryoelectronics and packaging provide concrete perspectives on how this goal can be tackled.

## Methods

### Electrostatic simulations and orbital splitting

For the calculation of the electrostatic potentials, we employed finite-element-method (FEM) simulations using COMSOL Multiphysics®. For each operational element, as shown in Figs. 2a, 3a, 4a of the main text, we solved Poisson's equation:

$$-\nabla(\epsilon(r)\nabla\Phi) = \frac{1}{\epsilon_0}\rho, \qquad (2)$$

with the electrostatic potential $\Phi$, charge density $\rho$, the dielectric constant of the sample $\epsilon(r)$ and the vacuum permittivity $\epsilon_0$. Dirichlet boundary conditions corresponding to the applied voltages were imposed at metallic gates. As the structure is intended to be filled with dilute electrons representing qubits whose behavior will be fully governed by the electrostatic potential in their absence, their charge was not included in $\rho$. We used the linearity of the model to simplify the variation of the applied voltages $V_i$ by calculating basis potentials $\Phi$ of each gate $i$ separately and combining the resulting total potential

$$\Phi = \sum_i V_i\Phi_i. \qquad (3)$$

Specifically, $\Phi_i$ is the potential for gate $i$ set to 1 V with all others at 0 V. This superposition approach is justified in regions where no or only very few electrons are present. In the IR zone, however, one needs to

**Table 1 | Dimensions of the micromagnets for the IR zone and manipulation zone, respectively, and associated magnetic field gradients**

| Case | dimension/nm³ | $\Delta B_\perp \lvert \partial B_\perp$ | $\Delta B_\parallel \lvert \partial B_\parallel$ |
| --- | --- | --- | --- |
| IR | 700 × 200 × 20 | <0.3 mT | 1.3 mT |
| SQG | 400 × 200 × 20 | 0.075 mT/nm | <0.01 mT/nm |
| TQG | 400 × 200 × 20 | 4 mT | 8.7 mT |

Quantities given in mT/nm refer to the derivative of the field at the assumed operation point denoted via $\partial$, quantities in mT to the total field difference between two qubits during the operation denoted via $\Delta$.

IR initialization and readout, SQT single-qubit gates, TQG two-qubit gates.

take the reservoir's contribution to $\rho$ into account. To do so, we first used the Thomas-Fermi approximation and solved the Poisson equation self-consistently, assuming a depleted two-dimensional electron gas (2DEG) in the channel and SET, for a specific gate voltage configuration that leads to the intended occupation of the reservoirs. To simplify fine-tuning of the gate voltages via the superposition approach, we subsequently modeled the reservoirs analogous to metallic gates, thus assuming perfect screening. The position of these gates was obtained from the region of nonzero charge density of the initial Thomas-Fermi solution. This neglects a change of the reservoir region in response to gate voltages and introduces a small error as the gradual screening by the 2DEG in the reservoirs is replaced by a hard boundary condition. As the reservoirs are relatively far from the region of interest, these approximations are compatible with our goal of demonstrating the feasibility to create an appropriate potential. The potential energies shown in the figures are referenced to the conduction band edge and given as $V = -e\Phi$.

For quantifying the effect of the variations in confinement in the T-junction (Fig. 2), we calculated the orbital splitting for the simulated potential of the quantum dot confining the qubit by solving the time-independent Schrödinger equation in two dimensions for each time step.

### Micromagnet design

Considering the requirements for the gate operations, we identified suitable dimensions for Cobalt micromagnets which provide the necessary field gradients. The resulting geometries and corresponding gradients are summarized in Table 1. Using thin layers ensures sufficient remanent magnetization when operating at low external magnetic fields, which we substantiated with OOMMF[59] simulations using material parameters from ref. [60]. Note that the perpendicular field gradient $\partial B_\perp$ for two-qubit gates arising from the magnet geometry is neither required nor harmful. As it is weaker than in the single-qubit zone and the gate duration is comparable, resulting relaxation errors are expected to be negligible.

The perpendicular field gradient for the IR zone, as well as the parallel gradient for the primary SQG position, are undesired and were rounded conservatively from simulations of potential misalignments during device fabrication. Here, a Gaussian misalignment of $\sigma_{xy} = 30$ nm in both horizontal directions was sampled. These gradients were then included in the dynamics model described in the following Methods section "Operation fidelities". A detrimental influence of micromagnets on other IR/SQG/TQG zones is negligible for the assumed spatial separations.

### Operation fidelities

To verify the feasibility of the architecture, we performed quantum dynamic simulations of each quantum operation using the simulation package qopt[44]. We calculated the quantum dynamics by solving the time-dependent Schrödinger equation for adequate model Hamiltonians to identify simple control pulses for initialization and readout, single-qubit gates and two-qubit gates. To extract meaningful fidelities, we included realistic noise values from past experiments[3,45,46]. All simulations were performed assuming the g-factor of Si.

For the initialization and readout procedure, we simulated a linear ramping pulse which converts between the $S(2, 0)$ and $\lvert \downarrow\uparrow\rangle$-state by sweeping the potential detuning $\epsilon$ of a double quantum dot adiabatically, besides a jump over the avoided $ST_-$-crossing. We utilized a Hamiltonian truncated to the relevant three-state basis of $\{\lvert T_0\rangle, \lvert S\rangle, \lvert T_-\rangle\}$,

$$H = \begin{pmatrix} 0 & \Delta B_\parallel/2 & 0 \\ \Delta B_\parallel/2 & -J(\varepsilon) & \Delta B_\perp/(2\sqrt{2}) \\ 0 & \Delta B_\perp/(2\sqrt{2}) & -B_\parallel \end{pmatrix}, \qquad (4)$$

taking the Zeeman splitting ($B_\parallel$), parallel ($\Delta B_\parallel$) and orthogonal ($\Delta B_\perp$) field differences between two dots spaced ~100 nm apart from micromagnet simulations and experimental data for the exchange energy $J(\varepsilon)$ from ref. [45]. Including further fast charge noise on the detuning $\varepsilon$ with a spectral density of $\sqrt{S_\varepsilon} = 0.02$ neV/$\sqrt{\text{Hz}}$ (adapted from ref. [45] assuming a gate lever arm of 0.1 eV/V) and optimizing a jump in $\epsilon$ at the avoided crossing of $\lvert T_-\rangle$ and $\lvert S\rangle$ induced by unintentional orthogonal field gradients gives target state fidelities exceeding 99.9% when choosing pulse lengths $t$ ~200 ns, fields of $B_\parallel \geq 20$ mT, $\Delta B_\parallel$ ~1 mT and a parasitic inter-dot orthogonal magnetic field difference $\Delta B_\perp \lessapprox 0.3$ mT. The separation of the electrons, which is well established, was assumed to occur perfectly adiabatically without thermal or dynamic excitation.

For single-qubit EDSR, spin and valley degree of freedom were considered in the Hamiltonian

$$H = \frac{1}{2}B_\parallel(x)\sigma_z + \frac{1}{2}B_\perp(x)\sigma_x + \Delta_{\text{VS},x}(x)\tau_x + \Delta_{\text{VS},y}(x)\tau_y \\ + \kappa_{\text{SVC},x}\tau_x \otimes \sigma_z + \kappa_{\text{SVC},y}\tau_y \otimes \sigma_z \qquad (5)$$

with $\sigma_i$ and $\tau_i$ denoting Pauli matrices on spin and valley space, respectively, $\Delta_{\text{VS}} = \Delta_{\text{VS},x} + i\Delta_{\text{VS},y}$ is a complex matrix element describing the coupling of the two lowest near-degenerate valley states in silicon[61] as a function of the electron position $x$ and $\kappa_{\text{SVC},i} = 0.01$ µeV parametrizes a g-factor variation between the valley states. EDSR-pulses as enveloped sinusoidal drives were then optimized for resonance frequency with the software framework qopt[44] and evaluated with respect to a process fidelity in the sense of ref. [62],

$$F_{\text{process}}(U) = \frac{1}{d_1^2}\lvert\text{tr}\{V^\dagger(U_t)_{\text{trunc}}\}\rvert^2, \qquad (6)$$

where $V$ describes the target gate unitary and $U_t$ the propagator in the eigenbasis of our model. The subscript *trunc* denotes truncation to the two-dimensional ($d_1$) spin subspace of the lower instantaneous valley state. This takes into account valley-leakage as valley excitations entail phase errors in subsequent operations[29]. From the simulations, we identified fast charge noise as the dominating noise contribution, which we modeled as an effective positional fluctuation with a spectral density of $\sqrt{S} = 0.1$ fm/$\sqrt{\text{Hz}}$ based on the experimental observation of $\sqrt{S(1\,\text{MHz})} = 0.2$ nV/$\sqrt{\text{Hz}}$ in ref. [45] transferred assuming a gate lever arm around 0.1 eV/V leading to assumed displacements of $\frac{\partial x}{\partial V} \sim 0.4$ nm/mV using calculations from ref. [3]. Taking into account the design choice of weaker magnetic gradients $\partial B_\perp$ ~0.1 mT/nm, our results indicate displacement amplitudes of around 20 nm (peak-to-peak) as a viable operation point to uphold a Rabi frequency near 10 MHz. This displacement amplitude constitutes a trade-off between achievable Rabi frequency and decoherence due to leakage on the valley space from potentially non-uniform valley splitting over the increased

**Fig. 6 | CNOT gate synthesis.** A CNOT gate is synthesized by a CZ gate, including Hadamard-like operations to account for necessary single-qubit operations (e.g., $\hat{z}$ rotations).

traveling distance of the electron during the pulse compared to previous experiments. For the simulations, we assumed that the qubit response scales approximately linearly with the driving field, which directly corresponds to the displacement amplitude. However, for conventional EDSR in a depletion-mode design, where a filled 2DEG is depleted to the few-electron regime, deviations from the linear scaling were observed for large driving amplitudes[3,17]. This deviation might have been caused by disorder, an anharmonic confinement potential, or driving via a valley dipole. If necessary, it can further be compensated by using stronger micromagnet gradients and smaller drive amplitudes. The possibility to reach a fidelity of 99.9% was found to be strongly correlated with the presence of a valley splitting $\gtrsim 30\,\mu$eV with sufficient spatial uniformity. Employing a model for $\Delta_{VS}$ incorporating alloy disorder effects on the valley splitting recently proposed in ref. [63] then yields >80% probability for fidelities >99.9% in the initial environment of the electron inside the manipulation zone under the assumption of $\langle E_{VS}\rangle = 2\langle|\Delta_{VS}|\rangle = 100\,\mu$eV, which is a conservative value compared to the current state of the art[64]. Adjustment of the electron position within the range of the manipulation zone, and therefore its valley environment, can further be utilized to circumvent spots with pathological behavior of the valley splitting that compromises the performance.

Two-qubit interaction was examined by coupling two single qubits described by Eq. (5) with a dot distance $d$ dependent exchange interaction term $\frac{1}{4}J(d)(\vec{\sigma}^{(1)}\cdot\vec{\sigma}^{(2)}-\mathbb{1})$, with $\vec{\sigma}^{(i)}$ being the Pauli matrices on both subspaces. An entangling interaction of CZ-class was simulated by optimizing towards $g_1 = 0$, $g_2 = 1$ of local invariants $g_1 = \frac{1}{16}\mathrm{tr}^2(m)\det(U^\dagger)$, $g_2 = \frac{1}{4}\left[\mathrm{tr}^2(m) - \mathrm{tr}(m^2)\right]\det(U^\dagger)$ according to ref. [65], with $m = U_B^T U_B$ and $U_B$ being the time evolution $U$ written in the Bell basis. The exchange energy $J(d)$ was calculated by solving the two-electron Schrödinger equation in one spatial dimension along the channel for each potential configuration of the shuttle pulse. $d$ was obtained as the separation between the minima of the double well potential in the two QuBus elements adjacent to the manipulation zone. Physically, both gate voltage fluctuations as well as charge noise contribute to fluctuations in $J$. Rather than modeling these independently, which is difficult to calibrate based on experiments anyway, we introduced an effective noise in $d$. The relevant infidelity contribution then arose from quasistatic position variations affecting the exchange interaction, which we rounded conservatively from ref. [3] to $\sigma_d = 10$ pm. This mathematical parametrization of $J$-noise in terms of position fluctuations can be expected to give a reasonable estimate because position variations directly translate to a change in the barrier height and width, which is the main factor for $J$ in the assumed barrier control mode. The assumed magnetic field gradient of $\partial B_\parallel$ -0.1 mT/nm was found suitable to realize entangling dynamics of CZ-class interaction on timescales of $t_{int}$ ~50 ns with a fidelity exceeding 99.9% conditional on coherent electron shuttling capabilities requiring sufficient ($\gtrsim 30\,\mu$eV) valley splitting.

### CNOT gate synthesis

CNOT gates are synthesized from CZ gates[11] (Fig. 6), since for electron-spin qubit platforms utilizing micromagnets, the natural choice for the implementation of CNOT-like two-qubit gates is the controlled-phase (CPHASE) gate. It requires a Zeeman energy difference $\Delta E_Z$ and an adiabatically switched exchange interaction $J(t)$ between two tunnel-coupled quantum dots[40–42]. The actual gate operation is based on

adiabatically turning on the exchange interaction $J(t)$, which shifts the energy levels of the antiparallel spin states in such a way that they acquire additional phases. Applying an exchange pulse for a duration $\tau = \pi\hbar/J$ combined with appropriately calibrated single-qubit gates[5,7,11,66] allows the implementation of a controlled-Z (CZ) gate or a CNOT gate to realize a universal gate set[11].

## Data availability
Electrostatic simulation results as shown in the figures including additional intermediate steps and gate fidelity simulation scripts have been deposited in the Zenodo database (https://doi.org/10.5281/zenodo.11110575).

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

## Acknowledgements

This work and all authors have been funded by the European Research Council (ERC) under the European Union's Horizon 2020 research and innovation program (Grant agreement No. 679342), by the German Research Foundation (DFG) under Germany's Excellence Strategy—Cluster of Excellence Matter and Light for Quantum Computing (ML4Q) EXC 2004/1-390534769, by the Federal Ministry of Education and Research (Germany), Funding reference number: 13N15652 and via Project Si-QuBus within the QuantERA ERA-NET Cofund in Quantum

Technologies implemented within the European Union's Horizon 2020 research and innovation program.

## Author contributions

M.K., A.W., I.S., L.R.S., and H.B. conceived the device layouts. M.K., A.W., H.Bh., M.B., E.K., I.S., and R.X. implemented and carried out electrostatic simulations. M.O., C.G., and J.D.T. carried out the dynamics simulations for each quantum operation. M.K., A.W., and H.B. conceived the operating strategy. M.K., A.W., R.O., and H.B. developed the wiring concept. H.B. and L.R.S. provided guidance to all authors. M.K., A.W., M.O., J.D.T., and H.B. wrote the manuscript.

## Funding

## Competing interests

The conveyor-mode shuttling device QuBus is covered by a patent family (EP4031486, US 2022/0293846 A1, CN114424346 A) by the work of the inventors M.K., I.S., H.B., L.R.S. and the patent application, co-owned by RWTH Aachen University and Forschungszentrum Jülich GmbH, is currently pending. The IR zone and a method for operating it for initialization is covered by a patent family (EP4031489, US 2023/0006669 A1, CN114402441) by the work of the inventors H.B., L.R.S., M.K. and the patent application, co-owned by RWTH Aachen University and Forschungszentrum Jülich GmbH, is currently pending. A method for operating the IR zone for readout is covered by a patent family (EP4031487, US 2022/0327072, CN114424344) by the work of the inventors H.B., L.R.S., M.K. and the patent application, co-owned by RWTH Aachen University and Forschungszentrum Jülich GmbH, is currently pending. The T-junction is covered by a patent family (EP4031490, US 2022/0344565, CN114514618) by the work of the inventors H.B., L.R.S., M.K. and the patent application, co-owned by RWTH Aachen University and Forschungszentrum Jülich GmbH, is currently pending. The manipulation zone is covered by a patent family (EP4031488, US 2022/0414516 A1, CN114402440) by the work of the inventors H.B., L.R.S., M.K. and the patent application, co-owned by RWTH Aachen University and Forschungszentrum Jülich GmbH, is currently pending. The quantum layer consisting of tileable unit cells is covered by a patent family (EP4031491, US 2022/0335322 A1, CN114424345) by the work of the inventors H.B., L.R.S., M.K. and the patent application, co-owned by RWTH Aachen University and Forschungszentrum Jülich GmbH, is currently pending. A method for distributing qubits between unit cells is covered by a patent family (WO2023/117064 A1) by the work of the inventors M.K., Cerfontaine and the patent application, owned by RWTH Aachen University, is currently pending in the designated PCT-states. A method for occupying unit cells with more than one qubit is covered by a patent family (WO2023/117065 A1) by the work of the inventors L.R.S., M.K., H.B. and the patent application, co-owned by RWTH Aachen University and Forschungszentrum Jülich GmbH, is currently pending in the designated PCT-states. The modified T-junction with a segmentation of the outer screening gate to allow a local pulsing is covered by a patent family (PCT/EP2023/055060) by the work of the inventors L.R.S., A.W., H.Bh., R.X., M.K., H.B., E.K., and the patent application, co-owned by RWTH Aachen University and Forschungszentrum Jülich GmbH, is currently pending in the designated PCT-states. A method for avoiding problematic spots in the shuttling channel by shifting the shuttling path laterally is covered by a patent family (PCT/EP2023/055058) by the work of the inventors Klos, M.O., M.K., H.B., J.D.T., and the patent application, co-owned by RWTH Aachen University and Forschungszentrum Jülich GmbH, is currently pending in the designated PCT-states. The modified QuBus device, including a global top gate, is covered by a patent family (PCT/EP2023/055061) by the work of the inventors L.R.S., Focke, I.S., H.B., and the patent application, owned by RWTH Aachen University, is currently pending in the designated PCT-states. The broadband phononic Bragg reflector is covered by a patent family (DE102021123046 B3, WO2023/031477 A1, CN117917211 A) by the work of the inventor H.B., and the patent application, owned by Forschungszentrum Jülich GmbH, is currently pending. L.R.S. and H.B. are founders and shareholders of ARQUE Systems GmbH. The remaining authors declare no competing interests.
