## [Peer Review File · Nature Communications]

REVIEWER COMMENTS

Reviewer #1 (Remarks to the Author):

The manuscript proposes a SpinBus architecture, where the quantum link consists of a conveyor-mode channel. The same channel gates are used to realize single- and two-qubit operations by dividing the channel into dedicated zones. These zones allow separation of the qubit operation, which is proposed to reduce crosstalk compared to having a full qubit manipulation in dense arrays. The proposal considers operation time scales within the coherence time of Si devices. The number of working qubits is estimated as a linear scale of 144 per 1000 coaxial cables.

This proposal manuscript has referenced previous architecture proposals from Refs. 17-21 and newly addresses a concrete structure for the shuttling channels of micron order, which includes a recently demonstrated conveyor-mode shuttling protocol. The manuscript addresses the small and fixed number of individual voltages gates needed to operate per channel, and such that the qubits can be spatially separated to create open space for wire routing. The qubit operation using the same proposed gate electrodes as an alternative approach is an interesting one. Although, these suggestions are not sufficiently addressed to their main aim and such presented claims themselves are not as effective as compared to the previous studies they refer to. Thus, prohibiting recommendation to Nature Communications. A more specialized journal for quantum information or applied physics seems more efficient in terms of being referenced by future works simply incorporating periodic gate structures.

Below are main comments that need to be addressed for the authors' major claims to be valid:

1. The "fan-out problem" is much addressed in other studies as in Ref. 21 where the aim is to move away from linear scaling but to achieve \sqrt{N} gates for N qubits which directly look into the million-qubit architecture. How does the newly proposed architecture address this problem as raised in the first part of the introduction? The claim of fan-out and the example number using coax lines in this manuscript does state a lowering of the density of gates, but this is different from the requirement to reduce the total number of gates from the millions of lines for million qubits that they state necessary for scaling.

2. In this sense, "scaling" in the Title is simply about lowering the density per area to allow control of certain number of qubits, 144 qubits using 1000 coaxes in the manuscript. Therefore, realizing a NISQ or surface code architecture does not seem to be the direct consequence, but it may rather be fair to compare with the allowed qubit numbers for the present dense architectures. This is not that electron shuttling is invalid for millions of qubits, but rather that the effectiveness of electron shuttling in the title should be properly claimed for the readers who look into the real advantages for making it sparse and difficulties remaining for million qubits that they should face.

3. The single- and two-qubit operations proposed by using the shuttling gates is yet to be proven practical. The claim of feasibility is not sufficiently addressed. Especially, shuttling gates that apply orders of magnitude larger microwave frequency needed for resonant spin manipulation cannot be imagined for such large overlapping structure. In addition, how is this architecture different from adding a single manipulation gate at the manipulation zone? How is the two-qubit gate feasibly pulsed at the end of a slow sinusoidal shuttling movement? How much crosstalk problem is mitigated compared to additional studies required for these newly proposed qubit manipulations? The claims should be backed up with realizable facts.

4. Throughout the manuscript, the validation of fidelities rely on operation speed vs. coherence time, which is already not a suitable measure for practical implementation as crosstalk, smoothness and confinement of potentials are not included in the validation. Otherwise they are claimed to be negligible but these quantitative discussions, compared to the 99.9% they claim, were not found in the manuscript.

The following comments are also suggested for non-specialized readers:

1. In the abstract, the authors informed that control using room temperature instruments can plausibly support at least 144 qubits, but much larger numbers are conceivable with cryogenic control circuits. Please specify the exact cryogenic temperature that will be used to control the qubits in the specified SpinBus architecture.

2. In Fig.2 (a), the authors indicated that straight shuttling is shown by the red path whereas the corner shuttling is shown by the blue path. However, I have observed that the corner shuttling has been shown by the green path instead of blue. Verify the color representation.

3. In Fig 1b, the authors indicated that the occupied potential minimum is indicated by a red circle, but this red circle is not visible in Fig 1b.

4. The authors informed that during straight shuttling, the orbital splitting equals or exceeds the threshold at all times (Fig. 2c). During corner shuttling, the orbital splitting drops to about 0.6 meV, slightly below the conservative target range (Fig. 2d). Provide a technical explanation for why the orbital splitting exceeds during straight shuttling and drops during corner shuttling.

5. Include a pictorial representation of the input terminals proposed for the T-junction and manipulation zones in the section discussing wiring complexity and solutions within the supplementary material.

6. The authors cited references for many parameters and formulas that were used for simulation. It will be helpful to the readers if you include them in the manuscript or supplementary material.

7. In methods M3, the authors informed that they included realistic noise values from past experiments to extract meaningful fidelities 3,42,43. Even though the realistic noise values used in the manuscript are indicated in the references, including them in the manuscript or supplementary material would be useful for readers.

8. Show the graph of fidelity while varying the pulse length and the presence of valley splitting, etc.

9.The authors compared their proposed work with a four-way junction. Cite the references for this four-way junction.

10.The authors expect that if needed, the orbital splitting can be increased by some combination of optimizing the geometry and pulse shapes, increasing the gate voltages, and dynamically adjusting the screening gate voltage. Provide some examples of geometry that could be effective in increasing orbital splitting.

Reviewer #2 (Remarks to the Author):

Reviewer #3 (Remarks to the Author):

In their manuscript the authors propose an architecture for scaling spin-based quantum processors in Si/SiGe heterostructures to larger qubit numbers. The proposal relies on a so-called electron conveyor which physically moves electrons in a sliding electric potential generated by gates connected to shared inputs. While there is a growing number of theoretical and experimental publications already discussing the electron conveyor – involving many of the present authors – the manuscript introduces the novel concepts of a T-junction and a manipulation zone. This allows for constructing a chip layout with two-dimensional connectivity where the qubits are shuttled to dedicated manipulation and readout zones on demand, while they are idling far from the detrimental effects of inhomogeneous magnetic fields of the manipulation zones.

The authors design a unit cell containing one manipulation and initialization / readout zone each and a number of qubits stored in the shuttling lanes. The integration of these individual components with the electron conveyor is discussed in detail and the authors simulate the error probabilities of the different operations. The discussion of initialization, readout and the shuttling in a straight line is based on both previous experiments and theory (Refs. [28-31] of the manuscript), while the discussion of the T-junction and the manipulation zone is based on numerical simulations presented in the manuscript. Finally, the authors provide a blueprint how an error-correcting code can be implemented on their architecture and discuss the complexity of the input signals, concluding that

up to 144 qubits (corresponding to 72 unit cells) can be realized with room temperature control electronics.

The idea presented in the manuscript addresses some of the major challenges currently faced by the spin qubit community: Both the short range of the native spin-spin interaction and fan-out of individually addressed voltage gates has hindered the development of large and highly connected devices so far. Spin shuttling as a possibility for alleviating these obstacles is not a new idea, however, the authors' work is innovative and original nonetheless as they present a full blueprint alongside a careful assessment based on current experimental capabilities. The design of the T-junction and the embedding of manipulation zones into the electron conveyor are decisive steps towards a functional demonstrator in the near future. The authors' claims are supported by sound simulations based on credible data obtained in previous experiments of the same group, with all relevant details being meticulously documented. Thus, the presented architecture should be taken serious as a realistic option for realizing spin-based quantum computing.

In conclusion I think that the manuscript is well-suited for publication in Nature Communications, both regarding the significance of the presented ideas as well as its scientific quality. The manuscript is well written, does not rely on expert knowledge of the reader to express its main points and the figures are quite clear. However, reading the manuscript left a number of minor questions which I feel should be clarified by the authors in order to strengthen the presentation of their results. These points are listed below as they appear in the manuscript. Once the authors addressed the following questions I am happy to recommend publication of their excellent and inspiring work in Nature Communications.

1. Zehner should be spelled Zener.

2. Ref. [31] is cited as a reference for motional narrowing. However, the experiment there was performed in natural silicon. What effect of the motional narrowing can be expected in isotopically purified host material which I presume is the authors go-to choice in the long term?

3. How much crosstalk can be expected between the legs of a T-junction? For example in Fig. 2a, what is the effect of operating the green/blue shuttling lane on the light red gate set of the perpendicular shuttling lane?

4. The authors claim that the orbital splitting during the corner shuttling can be increased by optimizing the pulse shapes. Further elaborating the protocol for transfer between the branches would be helpful for supporting this claim.

5. The intrinsic spin-orbit interaction of silicon, albeit weak, depends on the directions of motion and spin relative to the crystal axes. Can the authors justify the negligence of this effect in the simulation? Is there an ideal orientation of the bus lanes with respect to the crystal axes?

6. On page 5, the authors comment that single qubit gates may have a fidelity $> 99.9\%$ if the valley splitting “does not exhibit an exceptionally strong variation”. I feel this expression should be quantified.

7. In Fig. 3b there are two curves but only one of them is labeled in the legend.

8. The simulations of the gate fidelity are a core element of the manuscript, however, this part feels a tad weak. Especially, the requirements for the valley splitting seem somewhat arbitrary. I suggest the authors strengthen the presentation of their work by including a figure depicting the expected gate fidelity as a function of the valley splitting in the methods part M3, or some other form of more detailed illustration.

9. In Fig. 4 the two electrons undergoing a two-qubit gate are depicted red and blue (panel a) as well as orange (other panels). For clarity I suggest using a uniform depiction in all panels. Furthermore, the choice of colors in panel b is not ideal for readers with a red-green colorblindness.

10. While the two-dimensional connectivity is a great advancement for spin-based quantum computing, the number of input lines is still proportional to the number of qubits, which is not immediately convincing. I suggest the authors add some context to this point by comparing the scalability and perspectives on further development of their architecture to other proposals for long-range connectivity, e.g. the use of microwave resonators.

11. Perhaps I am missing something, but in Sec. M3, the quantities S_ϵ and S are not defined. Furthermore, ∂B and ΔB seem to be used interchangeably throughout the manuscript.

Reviewer #4 (Remarks to the Author):

The authors propose in their manuscript a generalised architecture for quantum computing with shuttling as the main mechanism for long distance interaction. The proposal works based on the conveyor belt method of shuttling where a small number of clavier gates is sufficient for long distance qubit transfer. The authors also describe clearly how they would manipulate the electrons across the 2D layout, especially where there are turning points at the T-Junctions. The authors also consider qubit manipulation during the initialization and readout process, as well as the operation of one- and two-qubit gates.

The authors have provided a satisfactory detailing of the architecture and showed an understanding of the challenges that needs to be overcome in this architecture. We would like to make the following recommendations for possible areas of improvement in the manuscript:

1. One of the key ideas that is used in this architecture seems to be that by keeping the magnetic field low (20--50 mT), it reduces the Zeeman energy difference between sites, and with valley splittings larger than 20 μeV , high fidelity shuttling can occur using the conveyor belt method. However, as the authors also described in parts of the paper, it is not guaranteed that the valley splitting would not go to less than 20 μeV or even zero. In these cases, there is a high likelihood of losing coherence. I understand that the authors mentioned briefly methods mitigating this in the supplementary. I would find it very useful for the authors to discuss this in the main text as well in greater detail about how exactly they plan to "adjust the shuttling path and manipulation position" as they describe in the supplementary material.
2. The architecture seems to work with very tight constraints on the qubit parameters and fidelities. Can the authors comment on how essential these constraints to the effectiveness of the architecture? A 99.9% fidelity is assumed for the transfer process and predicted for qubit SPAM, one-qubit, and two-qubit gates. How robust are these numbers against common sources of errors in qubit systems outside of charge noise, including but not limited to stray magnetic fields from the micromagnets in this architecture, as well as imperfections in device fabrication?
3. Do the authors have other studies, either from experiments or simulations, that confirms the small orbital splittings of 1-2 meV as quoted as a requirement for the smooth transfer of the electrons.

Overall, we believe that the topic explored in this manuscript is valuable for the community and we recommend the manuscript for publication, with the necessary recommendations described above.

Regards,

Kok Wai Chan

MengKe Feng

UNSW

Reviewer #5 (Remarks to the Author):

Point-by-point responses addressing the referees' comments

We have marked all revisions in our manuscript in red text color. The revisions reproduced in this document are in italic font.

Comments from referee #1

1. The "fan-out problem" is much addressed in other studies as in Ref. 21 where the aim is to move away from linear scaling but to achieve \sqrt{N} gates for N qubits which directly look into the million-qubit architecture. How does the newly proposed architecture address this problem as raised in the first part of the introduction? The claim of fan-out and the example number using coax lines in this manuscript does state a lowering of the density of gates, but this is different from the requirement to reduce the total number of gates from the millions of lines for million qubits that they state necessary for scaling.

We thank the referee for pointing out this need for clarification. Indeed, our proposal does not target a solution for operating millions of qubits with external controls. Rather, focusing on conveyor-mode shuttling primarily addresses the fan-out problem at the prefactor level by reducing the density of control lines. Increasing the qubit pitch significantly reduces the size requirements for subsequent wiring layers and eventually, integrated (cryogenic) control electronics. Another important aspect is the reduction of crosstalk, discussed in more detail below. Compared to dense architectures, it thus greatly improves the feasibility of integrating on the order of 100 qubits, a number competitive with present-day NISQ-era quantum processor sizes, in the short term. While this approach may appear limiting compared to crossbar approaches, it has the significant advantage that no futuristic assumptions regarding the ability to control many qubits with one pulse need to be made. We have clarified this distinction, in particular in the introduction of our manuscript as already stated above.

We note that our architecture additionally has a significant scaling potential if combined with cryoelectronic control approaches that involve local generation, modification or routing of control signals and most likely the local generation of DC bias voltages. Here, the main enabling factor is the space provided for such cryoelectronic circuits within each unit cell. The approach we deem most promising is to place the control circuits on a separate chip in direct geometric correspondence with each unit cell and connect them via flip chip integration. In combination with such control circuits, our architecture could overcome current fan-out problems completely. While it would be of great merit to discuss such approaches in more detail, the number of questions to be addressed would exceed a reasonable scope of the present manuscript. We have nevertheless

taken up the suggestion to outline the additional challenges that need to be overcome to achieve scalability to millions of qubits, and explicitly sketched ideas to do so using cryoelectronic control approaches in a dedicated section on pages 6-7 of the main text. While these largely remain to be demonstrated experimentally, the increased (and adjustable) footprint of the unit cell makes this approach much more viable than for dense arrays. In this sense, the SpinBus architecture also points to a rather concrete scaling pathway in a broader sense and addresses a key related challenge at the level of the quantum layer.

2. In this sense, "scaling" in the Title is simply about lowering the density per area to allow control of certain number of qubits, 144 qubits using 1000 coaxes in the manuscript. Therefore, realizing a NISQ or surface code architecture does not seem to be the direct consequence, but it may rather be fair to compare with the allowed qubit numbers for the present dense architectures. This is not that electron shuttling is invalid for millions of qubits, but rather that the effectiveness of electron shuttling in the title should be properly claimed for the readers who look into the real advantages for making it sparse and difficulties remaining for million qubits that they should face.

Following on from our response to the previous comment, the word "scaling" in the title is on the one hand to be interpreted in the sense of significantly increasing the qubit number compared to the *current* state of the art, rather than an asymptotic scaling relation without any bounds. On the other hand we do expect full scalability to be enabled by implementing cryogenic control electronics. We appreciate that this sense of scaling should be explained more clearly and have done so, as outlined in our response to the previous comment, in the introduction of the revised manuscript. Together with the abstract, we feel that this sufficiently clarifies the intended sense of the title. We would like to emphasize that the reduction of crosstalk is an important element, as it reflects a major impediment for high-fidelity control of dense arrays (see Ref.¹). We further argue that implementing more than 100 qubits with high-fidelity control would represent a very competitive NISQ-era realization. Regarding "compare with the allowed qubit numbers", we find it difficult to argue a well-defined bound, as the problems faced by current dense architectures do not lie primarily in increasing the number, but control errors and the tuneup of more complex devices.

3. (A) The single- and two-qubit operations proposed by using the shuttling gates is yet to be proven practical. The claim of feasibility is not sufficiently addressed. Especially, shuttling gates that apply orders of magnitude larger microwave frequency needed for resonant spin manipulation cannot be imagined for such large overlapping structure. (B) In addition, how is this architecture

different from adding a single manipulation gate at the manipulation zone? How is the two-qubit gate feasibly pulsed at the end of a slow sinusoidal shuttling movement? How much crosstalk problem is mitigated compared to additional studies required for these newly proposed qubit manipulations? The claims should be backed up with realizable facts.

To address point (A), we estimate the capacitance of a local shuttling structure (about ten microns in length and thus roughly a factor of 2 longer than what we aim for the manipulation gate sets) to be on the order of 10 fF. On a $50\ \Omega$ drive line, this corresponds to an RC-cutoff frequency of about 300 GHz, much higher than the typical or proposed spin qubit operating frequencies. Hence, we do not share the concern of the referee.

Regarding point (B) we do not claim a fundamental difference compared to using additional, individually connected manipulation gates. Using the shuttling gates is simply a matter of convenience to be economic regarding the number of signals. We have clarified this on page 4 of the revised manuscript:

«Two independent QuBus elements enable sufficient control over both detuning and tunnel coupling of a double quantum dot potential formed at the junction, thus eliminating the need for additional separately contacted gates.»

We also do not see any major hurdles in pulsing the control gates for two-qubit gates. Ideally one can use baseband control so that both shuttling and manipulation pulses can be generated with AWGs operating, e.g., around 2.5 GSa/s. Alternatively, high frequency components can be added with bias tees, making sure that transients decay before baseband components for two-qubit gates are applied. In either case, the desired exchange coupling strength is achieved by simply stopping the shuttling motion at the desired inter-qubit distance.

For crosstalk, two effects have to be considered: Control signals intended for one qubit directly coupling to another qubit on the chip and transmission of control signal between different lines. The latter requires a careful chip and PCB design with shielded transmission lines. Colless and Reilly² show that crosstalk levels as low as -40 dB can be achieved, which corresponds to control infidelities of order 10^{-4} even if both qubits have the same resonance frequency. One source for crosstalk within the quantum layer is the direct influence of voltages applied at other gate sets. In the manipulation zone this effect is strongest for the qubit closest to the junction, i.e., for the qubit below the micromagnet in the red gate set when the blue gate set is driven. Using the electrostatic model and a quadratic fit to estimate the dot position we calculate an unwanted displacement of

0.1 nm when applying a drive of ± 10 nm on the other qubit. This corresponds to a change in resonance frequency that is about a factor of $5 \cdot 10^{-3}$ smaller than the Rabi frequency, which, for a π -gate, induces an infidelity of roughly $6 \cdot 10^{-5}$. If needed, this crosstalk can be reduced either by increasing the distance to the qubit or by tailoring the applied pulses such that the fields cancel each other at the qubit position. We have included the estimation of crosstalk as clarification on page 5 of the revised manuscript:

« Regarding crosstalk, driving a single-qubit gate on a qubit right of the junction in Fig. 4b causes a relative electrostatic shift corresponding to 0.5 % of the driving amplitude for the other qubit located left of the junction (orange circle), 250 nm from the driven shuttling element. Conservatively assuming the same resonance frequency, this translates to an infidelity of approximately $6 \cdot 10^{-5}$ for a π -gate. For the more distant qubit in the right single-qubit manipulation region, crosstalk is even weaker. In addition, remaining crosstalk can be reduced further by specifically tailored pulses accounting for the respective opposite single-qubit operation. »

A second source of crosstalk intrinsic to the quantum layer is the charge coupling between two nearby qubits. The closest relevant distance will be that between two single-qubit manipulation zones, $\approx 0.6 \mu\text{m}$. We estimate that the displacement of one electron in response to the presence of another $0.6 \mu\text{m}$ away is less than 0.4 pm, which corresponds to a change in the resonance frequency that is about a factor $4 \cdot 10^{-5}$ smaller than the Rabi frequency and thus negligible. The effect of a spin state change will be even smaller.

4. Throughout the manuscript, the validation of fidelities rely on operation speed vs. coherence time, which is already not a suitable measure for practical implementation as crosstalk, smoothness and confinement of potentials are not included in the validation. Otherwise they are claimed to be negligible but these quantitative discussions, compared to the 99.9% they claim, were not found in the manuscript.

We agree that comparing the operating speed to the coherence time is not a good basis for estimating fidelities, but the models employed are more sophisticated, taking both quasistatic as well as fast charge noise and nuclear spin noise into account when simulating the complete qubit dynamics. We have ensured that the models are clearly outlined in the Methods section. For the two-qubit gate, confinement potentials are proxied by calculation of the exchange energy from electrostatics before simulating Hamiltonian dynamics.

A relevant assumption of these models for single-qubit manipulation is that the qubit response scales approximately linearly with the driving field on the scale of 10 nm displacement. There are indications in the literature that this may not be the case for conventional EDSR driving using one driving gate, possibly due to anharmonic confinement potentials, disorder and driving via valley states^{1,3}. If this becomes an issue, using stronger micromagnet gradients and a smaller drive amplitude can provide a solution for which single-qubit gate fidelities exceeding 99.9 % have been demonstrated. We have added a brief discussion of this possible issue to the Methods section of the manuscript on page 10:

«However, for conventional EDSR in a depletion-mode design, where a filled 2DEG is depleted to the few electron regime, deviations from the linear scaling were observed for large driving amplitudes^{1,3}. This deviation might have been caused by disorder, an anharmonic confinement potential, or driving via a valley dipole.»

5. In the abstract, the authors informed that control using room temperature instruments can plausibly support at least 144 qubits, but much larger numbers are conceivable with cryogenic control circuits. Please specify the exact cryogenic temperature that will be used to control the qubits in the specified SpinBus architecture.

This is an important question, which is however determined by factors that are not directly related to the quantum layer proposed here. Having said that, we favor a qubit temperature of 100 to a few hundred mK to ensure a minimal loss of gate fidelity, while operating electronics around 1.8 K. We recently patented an approach to achieve the required thermal insulation by using a phononic Bragg reflector and it will be subject to a separate publication⁴. The main idea is to use a layer stack of materials with a large acoustic impedance mismatch and a choice of the layer thickness to obtain a high reflectivity of phonons over the whole thermal spectrum. We now mention this approach as one attractive possibility on pages 7 of the manuscript:

«While we favor a qubit temperature on the order of 100 mK to ensure a minimal loss of gate fidelity, thermally isolating flip chip solutions may allow the operation of electronics at a higher temperature than the qubits. Working around 2 K would potentially make cooling powers at the level of Watts accessible. For this purpose, we propose the implementation of thermal insulation of the quantum layer from the electronics by a broadband phononic Bragg reflector⁵ to sustain a temperature gradient over a high-density interconnect solution. Simulations⁴ indicate that a heat

load below 1 mW/cm² can be achieved with a thickness compatible with high density vias with a micron-scale pitch. Using superconductors such as NbN or NbTiN with a critical temperature of a multiple of the operating temperature can lead to a very small heat transfer through the vias. »

6. In Fig.2 (a), the authors indicated that straight shuttling is shown by the red path whereas the corner shuttling is shown by the blue path. However, I have observed that the corner shuttling has been shown by the green path instead of blue. Verify the color representation.

We thank the referee for pointing out the inconsistency and have adjusted the colors accordingly.

7. In Fig 1b, the authors indicated that the occupied potential minimum is indicated by a red circle, but this red circle is not visible in Fig 1b.

We thank the referee for bringing the error to our attention and have corrected it.

8. The authors informed that during straight shuttling, the orbital splitting equals or exceeds the threshold at all times (Fig. 2c). During corner shuttling, the orbital splitting drops to about 0.6 meV, slightly below the conservative target range (Fig. 2d). Provide a technical explanation for why the orbital splitting exceeds during straight shuttling and drops during corner shuttling.

We appreciate that the shown drop in the orbital splitting during corner shuttling, caused by a temporarily reduced confinement, was suboptimal. The reduced confinement arises from the wide channel and the distance between the straight channel and the closest pocket location in the branching channel. We have since identified an improved and easy-to-implement protocol which employs a dynamic adjustment of one of the screening gate voltages to keep the orbital splitting within the target range. Pulsing the outer screening gate of the straight channel can move the qubit closer to the branching channel and increase the orbital splitting to about 1 meV. We have updated our manuscript accordingly on pages 2-3 of the revised manuscript:

«A drop in the orbital splitting during corner shuttling caused by the asymmetry of the gate layout at the junction which reduces confinement can safely be prevented (Fig. 2d) by dynamically pulsing the outer screening gate of the straight branch during transfer (Extended Data Fig. 1b). The pulse pushes the electron towards the branching channels, reduces the effect of the asymmetry and thus increases the confinement. »

9. Include a pictorial representation of the input terminals proposed for the T-junction and manipulation zones in the section discussing wiring complexity and solutions within the supplementary material.

We are not sure we correctly interpret the suggestion, but please note that no additional signal connections beyond the two involved conveyor lanes each driven with four signals are needed to operate the T-junction. For concreteness, we have attached an SEM micrograph of a prototype device (to be used in future experiments) in the appendix that also shows the wire routing, but since the details of the wiring is otherwise not discussed in the manuscript either, we favor not including this point to avoid inflating the manuscript.

10. The authors cited references for many parameters and formulas that were used for simulation. It will be helpful to the readers if you include them in the manuscript or supplementary material.

We appreciate that the referee's suggestion improves the accessibility of our manuscript. In the Methods section "Operation fidelities" on pages 9-10 of our revised manuscript, we therefore have additionally included and expanded on the formula used to calculate the fidelity and discuss the optimization metric used for the two-qubit gate in more detail. We also have improved the description of the exchange interaction Hamiltonian term. As for further parameters, we have ensured that relevant noise parameters used in the simulation are explicitly mentioned alongside their respective reference, as also detailed in our response to the next comment.

«[...] with respect to a process fidelity in the sense of Wood et al.⁶,

$$F_{\text{process}}(U) = \frac{1}{d_1^2} \left| \text{tr} \left\{ V^\dagger (U_t)_{\text{trunc}} \right\} \right|^2, \quad (1)$$

where V describes the target gate unitary and U_t the propagator in the eigenbasis of our model. The subscript *trunc* denotes truncation to the two-dimensional (d_1) spin subspace of the lower instantaneous valley state. »

«Two-qubit interaction was examined by coupling two single qubits described by Eq. (5) with a dot distance d dependent exchange interaction term $\frac{1}{4}J(d)(\vec{\sigma}^{(1)} \cdot \vec{\sigma}^{(2)} - \mathbb{1})$, with $\vec{\sigma}^{(i)}$ being the Pauli matrices on both subspaces. An entangling interaction of CZ-class was simulated by optimizing towards $g_1 = 0, g_2 = 1$ of local invariants $g_1 = \frac{1}{16} \text{tr}^2(m) \det(U^\dagger)$, $g_2 = \frac{1}{4} [\text{tr}^2(m) - \text{tr}(m^2)] \det(U^\dagger)$ according to Makhlin⁷, with $m = U_B^T U_B$ and U_B being the time evolution U written in the Bell basis. »

11. In methods M3, the authors informed that they included realistic noise values from past experiments to extract meaningful fidelities 3,42,43. Even though the realistic noise values used in the manuscript are indicated in the references, including them in the manuscript or supplementary material would be useful for readers.

We thank the referee for pointing out the need for additional clarification on the noise values. We have therefore ensured that the relevant noise values are numerically stated in the Methods section in the form in which they enter the simulated model. We have further complemented the description of the derivation from the measured values with the conversion factors used to estimate the final quantity.

«Including further fast charge noise on the detuning ε with with a spectral density of $\sqrt{S_\varepsilon} = 0.02 \text{ nV}/\sqrt{\text{Hz}}$ (adopted from Dial *et al.*⁸ assuming a gate lever arm of 0.1 eV/V) [...]»

«From the simulations we identified fast charge noise as the dominating noise contribution, which we modeled as an effective positional fluctuation with a spectral density of $\sqrt{S} = 0.1 \text{ fm}/\sqrt{\text{Hz}}$ based on experimental observation of $\sqrt{S(1 \text{ MHz})} = 0.2 \text{ nV}/\sqrt{\text{Hz}}$ in Dial *et al.*⁸ transferred assuming a gate lever arm around 0.1 eV/V leading to assumed displacements of $\frac{\partial x}{\partial V} \sim 0.4 \text{ nm/mV}$ using calculations from Yoneda *et al.*³»

12. Show the graph of fidelity while varying the pulse length and the presence of valley splitting, etc.

We appreciate that the valley splitting is likely the central quantity determining the feasibility of our proposal. We thus have appended Fig. 2 to this letter to better exemplify the obtained simulation results for the single-qubit manipulation. Fig. 2a shows simulated traces for the valley splitting matrix element $\vec{\Delta}$, displaying evolutions of 20 nm as needed to simulate our EDSR-driving drawn from a distribution of $\langle(|\Delta|)\rangle = 50 \mu\text{eV}$ as outlined by Wuetz *et al.*⁹. Fig. 2b shows correlations between the valley splitting and the fidelity, further elaborated on in the answer to referee #3, comment no. 8. In particular, Fig. 2c displays our results from the model shown in Methods section "Operation fidelities" for a single-qubit X-gate when varying the pulse length, with and without applying a white noise spectrum of $\sqrt{S} = 0.1 \text{ fm}/\sqrt{\text{Hz}}$ on the positional displacement during the EDSR-pulse, estimated from the experiments of Dial *et al.* and Yoneda *et al.*. Error bars represent the inner quartiles for a distribution of random valley splitting traces as depicted in (a). We identify a minimum around a driving amplitude of 10 nm, where the bulk of traces

reaches infidelities below 10^{-3} , corresponding to pulse lengths of 50 ns given the configuration of the magnetic field gradients as stated in Methods section "Micromagnet design". While these details are ultimately important, we favor not including them in the manuscript, which is intended to cover the minimum to underpin the feasibility of the architecture, while the underlying physics and exploration of the parameter space is better investigated separately. For manipulation, such details and possible ways to address them are covered in¹⁰. We attach another nearly completed manuscript addressing shuttling errors¹¹.

13. The authors compared their proposed work with a four-way junction. Cite the references for this four-way junction.

We thank the referee for pointing out the missing reference and have included Boter *et al.*¹² as reference on page 2 of the revised manuscript.

14. The authors expect that if needed, the orbital splitting can be increased by some combination of optimizing the geometry and pulse shapes, increasing the gate voltages, and dynamically adjusting the screening gate voltage. Provide some examples of geometry that could be effective in increasing orbital splitting.

As detailed in our response to comment no. 8, the improved protocol for corner shuttling employs a dynamically adjustment of the voltage applied to the outer screening gate of the straight channel. Avoiding an influence on other qubits stored in the channel requires a segmentation of the outer screening gate. We have included a comment on the adapted geometry on page 3 of the revised manuscript:

« To avoid any influence on other qubits stored in the straight branch, a segmentation of the outer screening gate at the junction can allow a local pulsing. »

Comments from referee #3

1. Zehner should be spelled Zener.

We thank the referee for the hint and corrected the spelling mistake in the manuscript.

2. Ref. [31] is cited as a reference for motional narrowing. However, the experiment there was performed in natural silicon. What effect of the motional narrowing can be expected in isotopically purified host material which I presume is the authors go-to choice in the long term?

This is indeed an important question. According to the analysis of Langrock *et al.*¹³, motional narrowing improves the spin dephasing time even in the absence of ²⁹Si isotopes due to either remaining ⁷³Ge isotopes or, if these are also removed, due to charge noise in the presence of magnetic field gradients. We have included a clarification and the reference to the analysis of in Langrock *et al.*¹³ on page 2 of the manuscript:

«The spin dephasing time of the shuttled electron spin is enhanced by motional narrowing, which contributes even in the absence of ²⁹Si isotopes due to remaining ⁷³Ge isotopes, and leads to a fidelity of approximately 99% for the transfer of a spin quantum state over a nominal shuttling distance of 560 nm¹⁴. In addition, motional narrowing is also expected for charge noise, albeit with a longer correlation length set roughly by the distance of the noise source from the channel¹³.»

3. How much crosstalk can be expected between the legs of a T-junction? For example in Fig. 2a, what is the effect of operating the green/blue shuttling lane on the light red gate set of the perpendicular shuttling lane?

We understand the referee's concern that storing qubits in the perpendicular branch directly at the junction while operating the straight branch may lead to crosstalk. If storing qubits in the perpendicular branch at least 100 nm away from the junction, any influence from operating the longitudinal branch can be avoided due to the rapid decay of electric fields in the heterostructure. We added a clarification on page 2 of the manuscript:

«Due to the rapid decay of electric fields, crosstalk is avoided by storing qubits in the perpendicular branch at least 100 nm away from the junction when operating the straight branch.»

4. The authors claim that the orbital splitting during the corner shuttling can be increased by optimizing the pulse shapes. Further elaborating the protocol for transfer between the branches would be helpful for supporting this claim.

As discussed in our replies to comments no. 4 and 14 from referee #1, we agree that the drop in orbital splitting during corner shuttling was not ideal and have since identified an easy-to-implement protocol which counteracts the loss of confinement during corner shuttling and thus always maintains an orbital splitting above 1 meV. We have included the details of the new protocol in our revised manuscript.

5. The intrinsic spin-orbit interaction of silicon, albeit weak, depends on the directions of motion and spin relative to the crystal axes. Can the authors justify the negligence of this effect in the simulation? Is there an ideal orientation of the bus lanes with respect to the crystal axes?

We share the referee's concern regarding the possibly detrimental effect of spin-orbit interaction (SOI) on spin coherence. However, due to its weak nature in SiGe, it can safely be neglected if the parameters for operating the SpinBus architecture are chosen properly: Langrock *et al.*¹³ concluded for SiGe that the SOI-induced error will stay below 10^{-4} for a shuttling distance of 10 μm with shuttling speeds slower than 100 m/s and the external magnetic field $B_{ext} > 5$ mT. Both conditions can be readily fulfilled in our proposed architecture. For manipulation, the expectation is that an intrinsic SOI is weak compared to the synthetic one arising from the micromagnet and will thus be compensated by the pulse calibration.

If a reduction of the effect of SOI is desirable, e.g., because a material system such as Ge/SiGe with higher intrinsic SOI is chosen, the geometry can indeed be optimized as the referee suggests. The SO term is given by $c(\alpha_{+[1\bar{1}0]}\sin(\Phi) + \alpha_{-[111]}\cos(\Phi))$ with Φ being the angle between $[1\bar{1}0]$ axis and B_{ext} ¹³. As the micromagnet's magnetization direction fixes the direction of B_{ext} , an optimization can be achieved in two possible way: First, if SOI during manipulation becomes problematic due to the higher shuttling speeds, the orientation of the shuttling lanes can be chosen in such a way that they are aligned with $[110]$ and $[1\bar{1}0]$ to minimize the contribution of SOI along the manipulation zone. Second, if the shuttling itself becomes problematic, an appropriate choice of the orientation of the shuttling lanes minimizes the overall SOI contribution. The alignment accuracy during fabrication of the quantum layer allows the implementation of both option.

We have added a brief comment on SOI on page 5 of the revised manuscript:

« *While the influence of spin-orbit interaction (SOI) during shuttling has been found to be minor¹³, it can safely be neglected for manipulation as the synthetic SOI used for EDSR is normally dominant.* »

6. On page 5, the authors comment that single qubit gates may have a fidelity $> 99.9\%$ if the valley splitting “does not exhibit an exceptionally strong variation”. I feel this expression should be quantified.

We agree that this can benefit from a quantification. We have added the concretization on page 6 that the valley splitting should « *exhibit integrated variations of less than $100\mu\text{eV}$ along the path* » of the EDSR movement, which is about 20 nm. See also our response to comment no. 12 from referee #1.

7. In Fig. 3b there are two curves but only one of them is labeled in the legend.

We thank the referee for pointing out this inconsistency and have added an entry for the potential to the legend.

8. The simulations of the gate fidelity are a core element of the manuscript, however, this part feels a tad weak. Especially, the requirements for the valley splitting seem somewhat arbitrary. I suggest the authors strengthen the presentation of their work by including a figure depicting the expected gate fidelity as a function of the valley splitting in the methods part M3, or some other form of more detailed illustration.

We thank the referee for pointing out the need to improve the discussion of requirements for the valley splitting. Since the detailed description of our architecture is the core of our manuscript and to provide a concise discussion of the simulated fidelities, we have improved the description of mitigation strategies of low valley splittings in the supplementary information and in particular referenced the study of Pazhedath *et al.*¹⁰, which covers a detailed analysis of error mechanisms for single-qubit gates and possible ways to reduce them:

« *The correlation length of the valley landscape is expected to be given by the spread of the electron’s wave functions ($\approx 20\text{nm}$), consistent with recent experiments¹⁵ that probe the valley splitting along a shuttling device. For the manipulation, simply stopping at a slightly different location in the roughly 200 nm wide region of favorable magnetic field gradient (see Fig. 4(b)) is thus very likely to avoid a problematic spot. For the proposed operating mode, valley-related single-qubit errors including their mitigation are discussed in detail in Ref.¹⁰. The shuttling path can*

be shifted laterally by applying voltages to the screening gates laterally defining the channel. A detailed study in Ref.¹¹ shows that, potentially in combination with slowing down by a factor three to five in remaining problematic spots, this can lead to transfer fidelities of 99.9% and higher over 10 μm with a very high probability. The voltage changes can be applied dynamically in synchronization with the electron motion. Alternatively, the screening gates can be split into individual, roughly micron-long sections to which different DC voltages are applied. The latter approach trades dynamical signals in favor of a larger number of DC voltages. »

In addition, we have appended Fig. 2 to this response letter to better exemplify the obtained simulation results for the single-qubit manipulation. Fig. 2a shows simulated traces for the valley splitting matrix element $\vec{\Delta}$, displaying evolutions of 20 nm as needed to simulate our EDSR-driving drawn from a distribution of $\langle(|\Delta|)\rangle = 50\mu\text{eV}$ as outlined by Wuetz *et al.*⁹. Using these values for the model described in Methods section "Operation fidelities", Fig. 2b then displays the correlation of the fidelity of an optimized single-qubit X -gate with parameters of the valley splitting trace. Each colored square represents one sample drawn for the valley splitting. From this, we draw the conclusion that, despite these parameters not fully defining all features of the valley splitting evolution, two criteria stick out, namely that: a minimal value (estimated $E_{\text{VS},\text{min}} = 2|\Delta| \sim 30\mu\text{eV}$) has to be fulfilled, and that variations along the path need to be small (declaring a rough estimate around $\delta E_{\text{VS}} < 100\mu\text{eV}$).

9. In Fig. 4 the two electrons undergoing a two-qubit gate are depicted red and blue (panel a) as well as orange (other panels). For clarity I suggest using a uniform depiction in all panels. Furthermore, the choice of colors in panel b is not ideal for readers with a red-green colorblindness. We are grateful for any suggestions that make our manuscript more accessible and have adjusted the colors accordingly so that they are more visible to colorblind readers. We have also adjusted the colors of the circles representing the electrons according to the referee's recommendation.

10. While the two-dimensional connectivity is a great advancement for spin-based quantum computing, the number of input lines is still proportional to the number of qubits, which is not immediately convincing. I suggest the authors add some context to this point by comparing the scalability and perspectives on further development of their architecture to other proposals for long-range connectivity, e.g. the use of microwave resonators.

We thank the referee for bringing to our attention, also in light of comments no. 1 and 2 from referee #1, that we should have defined more clearly how our proposal enables both scaling beyond the current state of the art using readily available technology, e.g., cryostat wiring and BEOL technology, and scaling to millions of qubits using cryogenic control approaches. We have added a clearer description of this important distinction in our revised manuscript.

11. Perhaps I am missing something, but in Sec. M3, the quantities S_ϵ and S are not defined. Furthermore, δB and ΔB seem to be used interchangeably throughout the manuscript.

We thank the referee for point out the lack of clarity and have updated our manuscript to include noise «*spectral density*» as the definition for the quantities \sqrt{S} . We have further revised the definitions of ΔB as absolute «*field differences between two qubits*» and ∂B as local «*derivatives*» to prevent misunderstandings.

Comments from referee #4

1. One of the key ideas that is used in this architecture seems to be that by keeping the magnetic field low (20–50 mT), it reduces the Zeeman energy difference between sites, and with valley splittings larger than $20\ \mu\text{eV}$, high fidelity shuttling can occur using the conveyor belt method. However, as the authors also described in parts of the paper, it is not guaranteed that the valley splitting would not go to less than $20\ \mu\text{eV}$ or even zero. In these cases, there is a high likelihood of losing coherence. I understand that the authors mentioned briefly methods mitigating this in the supplementary. I would find it very useful for the authors to discuss this in the main text as well in greater detail about how exactly they plan to “adjust the shuttling path and manipulation position” as they describe in the supplementary material.

We understand the need for further expansion on the aspect of mitigating small valley splittings during shuttling. We have expanded the supplementary section with further information on this aspect to complement the main text, outlining the conceptual approach to adjusting shuttling paths, e.g., by using DC offset voltages (see comment no.8 from referee #3). To retain brevity in the main part but still provide a detailed discussion, we have included the as yet unpublished work of Losert *et al.* as reference¹¹ and, with permission of the authors, have attached a draft version to this response letter. In that work, the concepts mentioned here only briefly are analyzed with respect to efficacy using simulations, showing that in principle, if the notion of valley splitting being dominated by alloy disorder holds true, there exist parameter sets that can mitigate these effects, potentially at the cost of wiring complexity and tuning effort. Furthermore, we have also referenced a study of Volmer *et al.*¹⁵ containing experimental data on valley landscapes and shifting electron trajectories with the screening gates:

«A subsequent study¹¹ as well as first experiments¹⁵ show that occasional lower values of the valley splitting can be avoided by laterally shifting the trajectory of the shuttled electron.»

«Lateral shifts of the shuttling path to avoid critical regions (see Supplementary information S1) can be achieved by antisymmetric changes of the voltages on the screening gates synchronized with the electron motion^{11,15}.»

2. The architecture seems to work with very tight constraints on the qubit parameters and fidelities. Can the authors comment on how essential these constraints to the effectiveness of the architecture? A 99.9 % fidelity is assumed for the transfer process and predicted for qubit SPAM, one-qubit, and two-qubit gates. How robust are these numbers against common sources of errors in qubit systems outside of charge noise, including but not limited to stray magnetic fields from the micromagnets in this architecture, as well as imperfections in device fabrication?

The philosophy in compiling the proposal was to provide concrete feasible numbers promising a good performance without compromising the accessibility by attempting to analyze the range of feasible values and possible trade-offs. For some values (e.g., magnetic field or field gradients), broad ranges are possible. Increasing the limiting noise amplitudes will of course directly affect gate fidelities. Generally, we find that the most crucial parameters is the distribution of valley splittings, as addressed in response to the previous comment. Regarding other aspects, the architectures does not pose very specific requirements and can be adapted to follow commonly chosen parameter values with demonstrated good performance approaching our predictions. To clarify the role of magnet variations and fabrication tolerances, we have added the following to the Methods section:

«The perpendicular field gradient for the IR zone as well as the parallel gradient for the primary SQG position are undesired and were rounded conservatively from simulations of potential misalignments during device fabrication. Here, a Gaussian misalignment of $\sigma_{xy} = 30$ nm in both horizontal directions was sampled. These gradients were then included in the dynamics model in following Methods section M3. A detrimental influence of micromagnets on other IR/SQG/TQG zones is negligible for the assumed spatial separations.»

3. Do the authors have other studies, either from experiments or simulations, that confirms the small orbital splittings of 1–2 meV as quoted as a requirement for the smooth transfer of the electrons.

This criterion is primarily derived from the study of Langrock, Krzywda *et al.*: the authors simulated the propagating potential in the QuBus for 800 ensembles of randomly distributed charged defects at the Si/SiO₂ interface. They concluded that for realistic device parameters, a splitting of the propagated potential into a double quantum dot can safely be prevented if a confinement corresponding to at least 1 meV can be achieved at all times during shuttling. One detailed experimental study is from Zajac *et al.*¹⁶, finding values of 3.0 ± 0.5 meV in nine different quantum dots.

The main conclusion of the simulations is that T-junctions are comparable to the regular shuttling sections. We are not aware of further references and believe that this question is addressed most effectively in more detail by experiments. We have added a brief comment on the criterion for the orbital splitting on page 2 of the manuscript:

«Langrock, Krzywda et al. determined that the required confinement strength to safely prevent the splitting of the shuttled potential minimum into a double quantum dot configuration in the presence of ensembles of randomly distributed charged defects at the Si/SiO₂ interface corresponds to an orbital splitting of at least 1 meV. This criterion is in agreement with experimentally obtained values typically found in static quantum dots¹⁶. »

APPENDIX

FIG. 1: SEM micrograph of a prototype of a T-junction.

FIG. 2: **Simulation results for single-qubit dynamics.** **a**, Valley splitting traces used for the simulations, see referee #1, comment no. 12 and referee #3, comment no. 8. **b**, Simulated EDSR fidelity as a function of valley splitting parameters, see referee #3, comment no. 8. **c**, Simulated fidelity as a function of EDSR amplitude, see referee #1, comment no. 12

REFERENCES

- ¹Undseth, B. *et al.* Nonlinear response and crosstalk of electrically driven silicon spin qubits. *Phys. Rev. Appl.* **19**, 044078 (2023).
- ²Colless, J. I. & Reilly, D. J. Cryogenic high-frequency readout and control platform for spin qubits. *Rev. Sci. Instrum.* **83**, 023902 (2012).
- ³Yoneda, J. *et al.* A quantum-dot spin qubit with coherence limited by charge noise and fidelity higher than 99.9%. *Nat. Nanotechnol.* **13**, 102–106 (2018).
- ⁴Duetz, D., Kock, S., Hangleiter, T. & Bluhm, H. Broadband phononic distributed Bragg reflector for thermally isolating integration of cryogenic control electronics with qubits. Manuscript in preparation.
- ⁵Bluhm, H. Isolator für kryoelektrische Chips bei extrem niedrigen Temperaturen unter 10K. Patent DE102021123046 (WO2023031477A1) (2021).
- ⁶Wood, C. J. & Gambetta, J. M. Quantification and characterization of leakage errors. *Phys. Rev. A* **97** (2018).
- ⁷Makhlin, Y. Nonlocal properties of two-qubit gates and mixed states, and the optimization of quantum computations. *Quantum Information Processing* **1**, 243–252 (2002).
- ⁸Dial, O. E. *et al.* Charge noise spectroscopy using coherent exchange oscillations in a singlet-triplet qubit. *Phys. Rev. Lett.* **110** (2013).
- ⁹Wuetz, B. P. *et al.* Atomic fluctuations lifting the energy degeneracy in Si/SiGe quantum dots. *Nat. Commun.* **13**, 7730 (2022).
- ¹⁰Pazhedath, A. M. *et al.* Large spin shuttling oscillations enabling high-fidelity single qubit gates. Preprint at <https://arxiv.org/abs/2403.00601> (2024).
- ¹¹Losert, M. P. *et al.* Simulation of spin shuttling in Si/SiGe heterostructures with random valley splitting due to alloy disorder. Manuscript in preparation.
- ¹²Boter, J. M. *et al.* Spiderweb array: A sparse spin-qubit array. *Phys. Rev. Appl.* **18**, 024053 (2022).
- ¹³Langrock, V. *et al.* Blueprint of a scalable spin qubit shuttle device for coherent mid-range qubit transfer in disordered Si/SiGe/SiO₂. *PRX Quantum* **4**, 020305 (2023).
- ¹⁴Struck, T. *et al.* Spin-EPR-pair separation by conveyor-mode single electron shuttling in Si/SiGe. *Nat. Commun.* **15**, 1325 (2024).
- ¹⁵Volmer, M. *et al.* Mapping of valley-splitting by conveyor-mode spin-coherent electron shut-

ting. Preprint at <https://arxiv.org/abs/2312.17694> (2023).

¹⁶Zajac, D. M., Hazard, T. M., Mi, X., Nielsen, E. & Petta, J. R. Scalable gate architecture for a one-dimensional array of semiconductor spin qubits. *Phys. Rev. Appl.* **6**, 054013 (2016).

REVIEWERS' COMMENTS

Reviewer #1 (Remarks to the Author):

The referee would like to express gratitude to the authors for their significant effort in revising the manuscript based on our comments and for providing additional information that helped further assessing the manuscript.

The main claim of the paper regarding NISQ computing has become effective, and the significance of this near-term implementation has been further emphasized.

With the inclusion of additional discussions addressing the concerns and allowed boundaries of heatload in the main text, the manuscript now appears ready to propose a concrete architecture and awaits demonstration.

Thus, the manuscript is now ready for publication in Nature Communications.

Reviewer #2 (Remarks to the Author):

Reviewer #4 (Remarks to the Author):

Thank you to the authors for providing a detailed reply to the the queries and making the relevant changes to the manuscript. In particular, we thank the authors for their consideration of the valley effects and also attaching the manuscript by Losert et al., which is a very interesting read into the methods that are being considered for mitigating valley effects in spin transport. I completely understand that the difficulties of interface steps, small valley splittings and valley phase flips are not problems that can be solved quickly or easily so I appreciate that the authors are taking this into account.

I also appreciate the philosophy in balancing rigour in the analysis and accessibility of the architectural design. I think the authors have provided sufficient additional description of their

approach. I am also satisfied with their justification of the small orbital splittings. I would just like to encourage the authors to consider the variability of these orbital splittings in future studies of this method, but that would not be required for this current study.

Overall, I believe that the manuscript is now in a good position, and I would like to strongly recommend the manuscript for publication.

Regards,

Kok Wai Chan

Mengke Feng

UNSW

Reviewer #5 (Remarks to the Author):
